



# Exploring climate stabilisation at different global warming levels in ACCESS-ESM-1.5

Andrew D. King[1,2], Tilo Ziehn[3], Matthew Chamberlain[4], Alexander R. Borowiak[1,2], Josephine R. Brown[1,2], Liam Cassidy[1,2], Andrea J. Dittus[5], Michael Grose[4], Nicola Maher[6,2], Seungmok Paik[7], Sarah E. Perkins-Kirkpatrick[8,2], Aditya Sengupta[1,2]

[1]School of Geography, Earth and Atmospheric Sciences, University of Melbourne, Parkville, Victoria, Australia.
[2]ARC Centre of Excellence for Climate Extremes, Australia
[3]CSIRO Environment, Aspendale, Victoria, Australia
[4]CSIRO Environment, Hobart, Tasmania, Australia
[5]National Centre for Atmospheric Science, Department of Meteorology, University of Reading, Reading, United Kingdom
[6]Research School of Earth Sciences, The Australian National University, Canberra, Australian Capital Territory, Australia
[7]Irreversible Climate Change Research Center, Yonsei University, Seoul, South Korea
[8]School of Science, UNSW Canberra, Canberra, Australian Capital Territory, Australia

*Correspondence to*: Andrew D. King (andrew.king@unimelb.edu.au)

**Abstract.** Under the Paris Agreement, signatory nations aim to keep global warming well below 2°C above pre-industrial levels and preferably below 1.5°C. This implicitly requires achieving net-zero or net-negative greenhouse gas emissions to ensure long-term global temperature stabilisation or reduction. Despite this requirement, there have been few analyses of stabilised climates and there is a lack of model experiments to address our need for understanding the implications of the Paris Agreement. Here, we describe a new set of experiments using the Australian Community Climate and Earth System Simulator earth system model (ACCESS-ESM-1.5) that enables analysis of climate evolution under net-zero emissions, and we present initial results. Seven 1000-year long simulations were run with global temperatures stabilising at levels in line with the Paris Agreement and at a range of higher global warming levels. We provide an overview of the experimental design and use these simulations to demonstrate the consequences of delayed attainment of global net-zero carbon dioxide emissions. As the climate stabilises under net-zero emissions, we identify significant and robust changes in temperature and precipitation patterns including continued Southern Ocean warming and reversal of transient mid-latitude drying trends. Regional climate changes under net-zero emissions differ greatly including contrasting trajectories of sea ice extent between the Arctic and Antarctic. We also examine the El Niño-Southern Oscillation (ENSO) and find evidence of reduced amplitude and frequency of ENSO events under climate stabilisation relative to projections under transient warming. An analysis at specific global warming levels shows significant regional changes continue for centuries after emissions cessation. Our findings suggest substantial long-term climate changes are possible even under net-zero emissions pathways. These simulations are available for use in the community and hopefully motivate further experiments and analyses based on other earth system models.



## 1 Introduction

The world has warmed by over 1°C above pre-industrial levels due to anthropogenic greenhouse gas emissions (IPCC, 2021; Haustein et al., 2017). Anthropogenic emissions of carbon dioxide have had major impacts on the climate already, and these impacts are worsening as the rate of emissions continues at record high levels (Friedlingstein et al., 2022). The growing impacts of climate change motivated the Paris Agreement of 2015 in which signatory nations agreed to aim to limit global warming to well below 2°C above pre-industrial levels and preferably to below 1.5°C. Limiting global warming in line with the Paris Agreement implicitly requires at least stabilising global temperatures if not cooling back towards pre-industrial conditions (Rogelj et al., 2017). Thus, it is imperative that humanity gains a greater understanding of the long-term consequences of low global warming levels and, in contrast, the implications of failing to meet the Paris Agreement through continued global warming or stabilisation at higher global warming levels.

Model analyses suggest that there is a near-linear relationship between cumulative carbon dioxide emissions and global-average temperature change (Seneviratne et al., 2016; IPCC, 2021; Allen et al., 2022). Furthermore, earth system model (ESM) simulations and simpler model runs performed as part of the Zero Emissions Commitment Model Intercomparison Project (ZECMIP; Jones et al., 2019) suggest that cessation of carbon dioxide emissions will result in an almost immediate halt to global warming and near-zero global temperature change for the following century (MacDougall et al., 2020; Palazzo Corner et al., 2023). These studies suggest that an emissions level very near zero is required to halt global warming in line with the Paris Agreement.

Since the Paris Agreement was signed, there has been a significant effort to understand what 1.5°C and 2°C global warming levels (GWLs) would entail in terms of climate change impacts, but this body of work has not focussed on net-zero emissions scenarios. This is primarily due to a lack of availability of suitable model simulations in the immediate aftermath of the Agreement being developed. Studies on the 1.5°C and 2°C GWLs were in part motivated by the urgent need for literature ahead of the Intergovernmental Panel on Climate Change (IPCC) Special Report on 1.5°C global warming (SR1.5; Masson-Delmotte et al., 2018). Work was undertaken to examine 1.5°C and 2°C GWLs, and rapidly inform SR1.5, using existing transient multi-model ensembles (e.g. King et al., 2017; Schleussner et al., 2016), pattern scaling (Tebaldi and Knutti, 2018; Seneviratne et al., 2016; King et al., 2018), single coupled model ensembles (Sanderson et al., 2017) and multi-model atmosphere-only ensembles (Mitchell et al., 2017). However, these methods are all, to some extent, based on transient climate states. The characteristics and relative merits of widely used methods for examining GWLs is detailed by James et al., (2017) with a brief summary provided here:

- Time sampling methods, based on extracting GWLs from existing simulations, generally use a combination of the Scenario Model Intercomparison Project (ScenarioMIP; O'Neill et al., 2016) projections with continued increases in greenhouse gas concentrations and global warming. Time sampling has become commonly used for examining





GWLs (King et al., 2017; Schleussner et al., 2016; Nangombe et al., 2018) and was the principal method employed in the IPCC Sixth Assessment Report Working Group 1 (IPCC, 2021). This method samples from continually

warming transient simulations.

- Pattern scaling also typically makes use of ScenarioMIP experiments (Tebaldi and Knutti, 2018; King et al., 2018; Tebaldi and Arblaster, 2014), although earlier studies used carbon dioxide-only forced simulations (Mitchell, 2003). The method involves identification of well understood climate changes expected under high greenhouse gas forcings and assumes a linear scaling with temperature changes to estimate patterns under lower greenhouse gas

forcing or GWLs. This method typically samples from continually warming simulations, but has been applied to slow-warming longer simulations previously (Mitchell, 2003).

- Bespoke simulations using the Community Earth System Model (Sanderson et al., 2017) achieve slow global temperature change through near-zero carbon dioxide emissions pathways in the last few decades of the 21st century. While these represent a climate nearer to stabilisation than the other methods described here, they are only

run to 2100 and thus have not had enough time for slow processes to continue and affect the global climate.

- Global atmosphere-only model simulations were developed as part of the "Half a degree additional warming, prognosis and projected impacts" (HAPPI) project (Mitchell et al., 2017, 2016). Simulations were run under warmed sea surface temperatures and increased greenhouse gas concentrations derived from coupled model differences between the recent climate and low-end ScenarioMIP simulations in the late 21st century. This

experimental setup is derived from transient climate data but based on slower changes than the time-sampling method typically uses.

There is a gap between the analyses previously performed to understand the implications of the Paris Agreement (based on transient climate states) and the intent of the Agreement itself (for a stabilised or cooling climate under net-zero or net-negative emissions). It is known that there are significant differences between transient and stabilised climates with respect to regional

temperatures (Manabe et al., 1991; King et al., 2020; Joshi et al., 2008), atmospheric circulation and precipitation (Grose and King, 2023; Sniderman et al., 2019; Ceppi et al., 2018), and ocean characteristics (Armour et al., 2016; Long et al., 2014), including sea level rise (Nauels et al., 2019). While examination of low GWLs in transient climate states is useful for understanding near-term climate changes, it is imperative that understanding of stabilised and overshoot climate behaviour, in line with the Paris Agreement goals, is improved. Projections based on transient methods may not be used as a proxy for

stabilised climates aligned with the Paris Agreement.

There are few long stabilised climate simulations that would be consistent with the intent of the Paris Agreement. While the tier 1 experiment in the ZECMIP protocol results in near-stable global-average temperatures, ESM simulations are initiated from the same cumulative emissions levels resulting in a range of GWLs, and the simulations are typically around 100 years in length (Jones et al., 2019). Ensembles of millennial-length simulations based on fixed concentration levels, such as




LongRunMIP (Rugenstein et al., 2019) or single-model experiments (Fabiano et al., 2023), warm continuously as fixed carbon dioxide concentrations are equivalent to a low continuous rate of carbon dioxide emissions. Again, these ensembles result in different amounts of warming between models dependent on individual models' equilibrium climate sensitivity.

There are also limited studies that explore stabilisation at different global warming levels. This is important as the temperature response after net-zero has been found to be dependent on total cumulative emissions (Jenkins et al., 2022; Allen et al., 2022),

and thus simulations must be run at multiple branching points to fully understand the post net-zero climate evolution. ZECMIP aims to achieve this, however, there are few branching points, and only a small set of models have been run with more than one branching point.

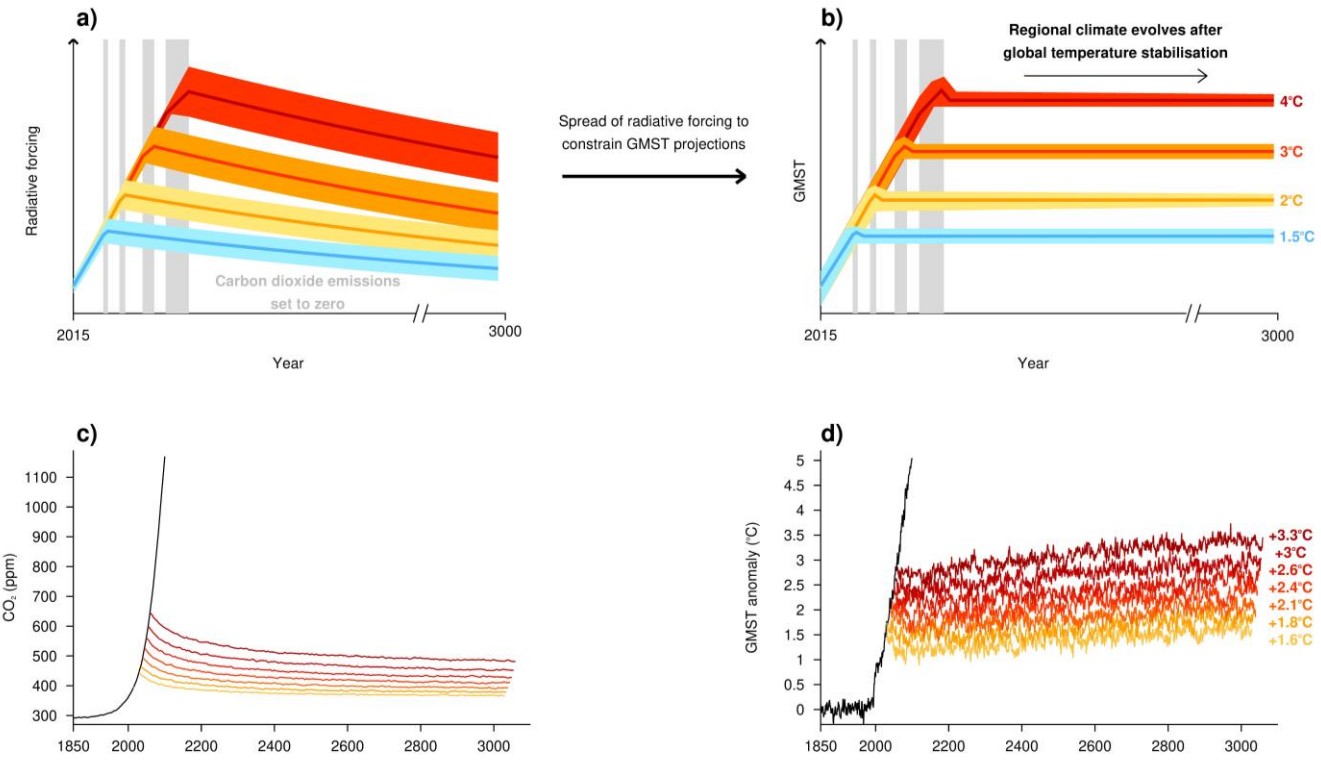

**Figure 1: a), b) Schematics of the proposed design of a multi-model ensemble of simulations at targeted global warming levels,**
**adapted from King, Sniderman, et al., (2021), and implemented with ACCESS-ESM-1.5 here. c) Atmospheric carbon dioxide (CO₂)**
**concentrations and d) global mean surface temperature (GMST) anomalies, relative to the 1850-1900 baseline, in the ACCESS-**
**ESM1-5 historical and SSP5-8.5 simulations (black) and the net-zero greenhouse gas emissions simulations (grading from yellow to**
**red for later emissions cessation). The net-zero emissions simulations begin in 2030, 2035, 2040, 2045, 2050, 2055 and 2060. The**
**global warming levels for the last 30 years of each of the net-zero emissions simulations are shown in d).**

Given the knowledge gap described above, new model simulations have been proposed to further address the needs of policymakers responding to the Paris Agreement (King et al., 2021a). In this framework, model simulations are initiated from different points in the high greenhouse gas emissions scenario, Shared Socioeconomic Pathway (SSP)5-8.5 and run forward

in emissions-driven mode with net-zero carbon dioxide emissions and 1850-level non-$CO_2$ greenhouse gases and aerosol concentrations (Figure 1 a,b). These simulations would be run for several centuries with the starting point chosen in each ESM

so that the simulations stabilise near targeted GWLs, including the 1.5°C and 2°C GWLs referred to in the Paris Agreement. Simulations in carbon emissions-driven mode more easily correspond to policy targets and offer other advantages relative to concentration-driven projections (Sanderson et al., 2023).

We must note that stabilised simulations are unlikely to represent plausible future scenarios, since it is unlikely that we would stabilise at net-zero without then immediately going to net-negative emissions (including carbon dioxide removal). However,

stabilised simulations are a useful baseline and reference for understanding stabilised climates before then exploring different plausible scenarios of net negative emissions.

Here, we describe a set of seven 1000-year simulations that follows the framework of King, Sniderman, et al., (2021) and aims to address the current knowledge gap around the global and regional climate response to stabilisation. These simulations were run using the Australian Community Climate and Earth System Simulator (ACCESS) ESM version 1.5 (Ziehn et al., 2020)

described in section 2.1 and the experiments are described in section 2.2. In section 2.3 we describe our analysis methods. In section 3.1, we discuss the results based on temporal evolution of the simulations and in section 3.2 we examine projections through the framework of GWLs. There are many applications of these simulations and here we perform a preliminary examination of climate extremes in section 3.3 and the El Niño-Southern Oscillation (ENSO) in section 3.4. In section 4 we provide a summary and conclusions.

## 2 Data and Methods

### 2.1 ACCESS-ESM-1.5

The ACCESS-ESM-1.5 model is Australia's latest generation ESM and is a participant model in the sixth phase of the Coupled Model Intercomparison Project (CMIP6; Eyring et al., 2016). It is comprised of the Unified Model version 7.3 (Martin et al., 2010), the Community Atmosphere Biosphere Land Exchange model (CABLE; Wang et al., 2011), the Modular Ocean Model

version 5 (Griffies, 2012) with Whole Ocean Model of Biogeochemistry and Trophic-dynamics (WOMBAT; Oke et al., 2013), the CICE sea ice model version 4.1 (Hunke and Lipscomb, 2010), and it uses the OASIS-MCT coupler (Craig et al., 2017). The ACCESS-ESM-1.5 model has an atmospheric grid spacing of 1.875° longitude by 1.25° latitude and a variable ocean grid with coarsest spacing of 1° by 1°. ACCESS-ESM-1.5 exhibits small biases in radiative flux terms, atmospheric and oceanic properties and carbon cycle characteristics (Ziehn et al., 2020). Further details on the configuration and basic model evaluation

may be found in Ziehn et al., (2020).





ACCESS-ESM-1.5 has an equilibrium climate sensitivity of 3.87°C and a transient climate response of 1.95°C (Ziehn et al., 2020), both of which are well within the range of CMIP6 ESMs (IPCC, 2021) and within estimated likely ranges from a recent community assessment (Sherwood et al., 2020). Overall, ACCESS-ESM-1.5 evaluates favourably compared with many other CMIP6 models for many important climate system properties, including for recent mean climate conditions (Xu et al., 2021),

some ENSO characteristics (Planton et al., 2021) and extremes indices (Kim et al., 2020). The model also exhibits unusually small drift in its pre-industrial control simulation (Irving et al., 2020), so drift is less likely a factor in explaining the projected climate evolution after emissions cessation than in other ESMs. The model performs well in relation to Australian climate and produces a particularly dry projection for the region, so has been selected for downscaling and detailed analysis for national projections (Grose et al., 2023; Rashid et al., 2022). As with all CMIP6 ESMs, it has shortcomings and deficiencies, for

example representation of different ENSO flavours appears to be worse in ACCESS-ESM-1.5 than in other CMIP6 models (Hou & Tang, 2022).

## 2.2 Experiments

For this particular analysis, the ACCESS-ESM-1.5 model was run in emissions-driven mode with net-zero carbon dioxide emissions and 1850 levels of other greenhouse gases and anthropogenic aerosols for 1000 years, following King, Sniderman,

et al., (2021). These simulations were initialised from seven different timestamps in the esm-SSP5-8.5 r10i1p1f1 simulation: 2030, 2035, 2040, 2045, 2050, 2055 and 2060. These simulations are hereafter referred to as "NZ" and the year of emissions cessation (e.g. NZ2030). The net-zero simulations exhibit substantial reductions in atmospheric carbon dioxide concentrations (Figure 1c) due to uptake by the land and ocean, similar to findings based on ZECMIP ESMs (MacDougall et al., 2020). The global mean surface temperature (GMST) slightly decreases in the first 20-50 years after the rapid shift to net-zero emissions

and reduced non-CO2 and aerosol concentrations has taken place. Much of the analysis described in section 2.3 is conducted on later periods to avoid the effect of this rapid change. GMST then slowly increases over the remainder of these simulations at a rate of around 0.03-0.05°C per century (Figure 1d) which is about one-fortieth of the current rate of observed global warming. The lack of long-term global cooling despite reduced atmospheric carbon dioxide concentrations is primarily due to slow ocean processes (MacDougall et al., 2022; Armour et al., 2016). The rate of global warming in these ACCESS-ESM-1.5

simulations is a slower rate of global temperature change than would be achieved with fixed atmospheric carbon dioxide concentrations (Rugenstein et al., 2019), but is slightly higher than could be achieved within the adaptive emission reduction approach framework where emissions are allowed to vary with the goal of achieving near-zero trends in global mean surface temperature (Terhaar et al., 2022).

These ACCESS-ESM-1.5 simulations were analysed and compared against an ensemble of 40 historical and corresponding

concentration-driven SSP5-8.5 simulations run with the same model. There is little difference between the emissions-driven and concentration-driven SSP5-8.5 simulations but a large ensemble exists for the concentration-driven version of the model.



This large ensemble of transient simulations and the seven quasi-stabilised 1000-year long net-zero emissions simulations allow for robust comparison of climate states as a function of both time and GWL.

## 2.3 Analysis

### 2.3.1 Global and local changes

First, global and large-scale annual-average characteristics of the stabilised runs were computed. GMST and global annual-average precipitation were calculated for the stabilised and transient ensembles. Also, land and ocean characteristics were examined separately. Global annual-average land and sea surface temperatures were computed and the difference taken as this is a commonly used climate change metric (e.g. Braganza et al., 2003). Global annual-average temperatures through the entire depth of the ocean were calculated to better understand the inertia of the ocean in response to rapid emissions followed by emissions cessation. Additionally, March and September sea ice extent in the Arctic and Antarctic regions was computed to examine seasonally extreme sea ice extent evolution in the post-net-zero runs relative to the transient case. A threshold of one million square kilometres was used to define virtually sea ice free conditions following Screen & Williamson, (2017). Several of the metrics analysed exhibit very high interannual variability, including the land-ocean temperature difference, average precipitation rate and sea ice extent metrics, so moving decadal averages of these indices are plotted.

The evolution of surface temperatures across the planet under net-zero emissions was explored by computing the difference in annual-average temperatures between the years 800-999 and 200-399 after emissions cessation in each of the 1000-year simulations. These are two long 200-year periods chosen to represent quicker and slower responses to emissions cessation and with the first period well after the rapid change in forcings was imposed to the simulations. Temperature difference values were averaged across the seven simulations but were similar between simulations (Figure A1). The long-term effect of delays to emissions cessation was examined by calculating the difference between the millennium-length simulations in the years 800-999 between adjacent simulations (i.e. NZ2035 minus NZ2030, NZ2040 minus NZ2035, etc.). The average effect of a 5-year delay was calculated, although the effect did not differ greatly between each pair of simulations (Figure A2).

The local timing of peak temperatures relative to the point of emissions cessation was examined to better understand the evolution of local climates under net-zero emissions pathways. The timing of peak 11-year annual-average temperatures at each location is identified for concatenated historical, esm-SSP5-8.5 and 1000-year long net-zero emissions simulations. This was repeated for each concatenated series based on esm-SSP5-8.5 up to the branching point of each of the seven 1000-year simulations, and the timing relative to emissions cessation was averaged and plotted.

The changing pattern of temperature change after emissions cessation relative to a transient warming climate was also investigated. The 1850-1900 period averaged across the 40 ACCESS-ESM-1.5 historical simulations was used as a proxy for pre-industrial conditions. Note, this is consistent with the IPCC Sixth Assessment Report (IPCC, 2021), but previous work



suggests this is around 0.1°C warmer than estimates of 18th century temperatures representative of earlier in the Industrial Revolution (Schurer et al., 2017; Hawkins et al., 2017). The local warming relative to global-average warming was computed in the 40 SSP5-8.5 simulations for 2030-2069. Local warming relative to global warming was also computed in the years 200-
399 and 800-999 after emissions cessation in the millennial-length simulations. The difference in these warming patterns with the transient pattern based on SSP5-8.5 was calculated.

### 2.3.2 Global warming level projections

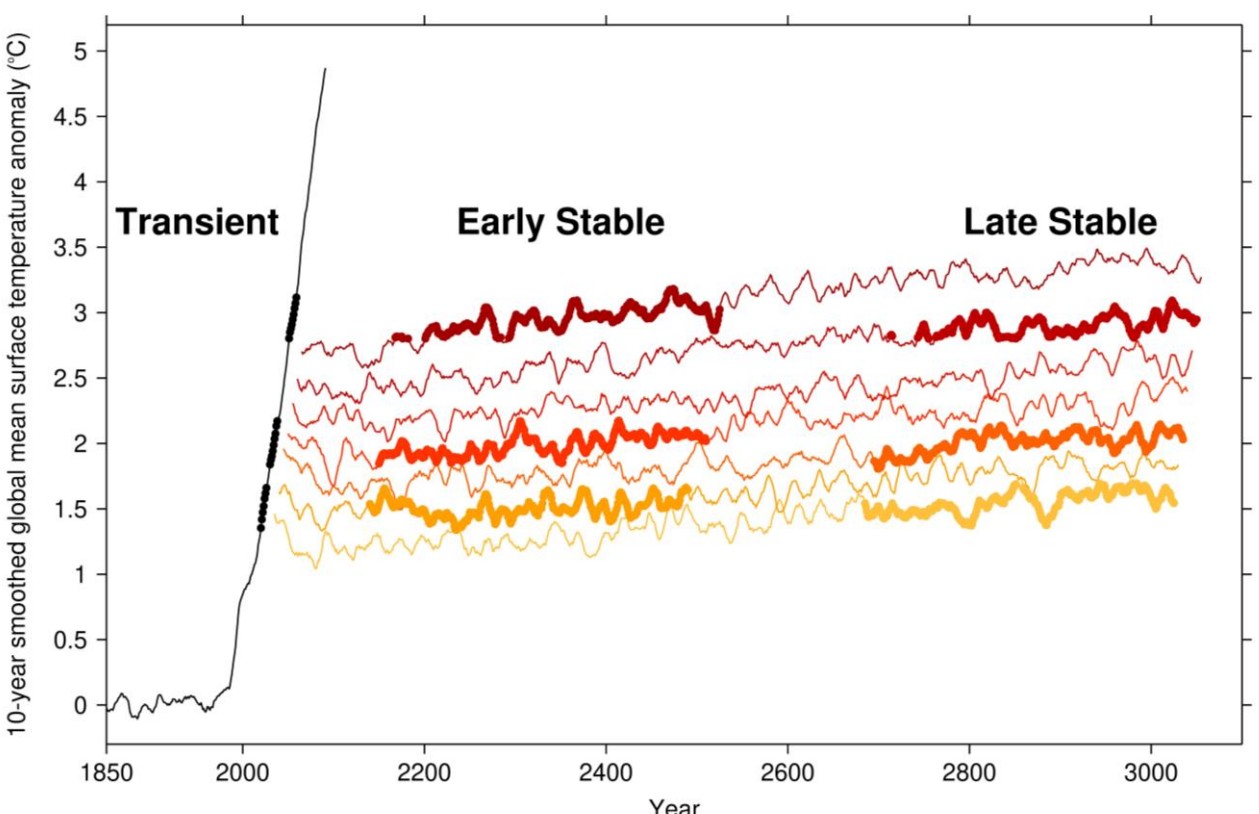

Figure 2. Figure showing the extracted periods of 1.5°C, 2°C and 3°C global warming levels in the transient (black) and net-zero
emissions (yellow to red) simulations. GWLs are extracted as all years within decades of global-average temperature of 1.5°C, 2°C and 3°C ± 0.2°C in the transient simulations and net-zero emissions simulations. Extracted decades are shown in bold. An 1850-1900 baseline is used as a proxy for a pre-industrial climate. Only one transient case is shown here for illustrative purposes, but all 40 concentration-driven SSP5-8.5 ensemble members are used.

In addition, analysis based on GWLs was conducted in the transient SSP5-8.5 and stabilised 1000-year long simulations (Figure 2). Transient 1.5°C, 2°C and 3°C GWLs were defined by extracting all years within decades where the average temperature in the SSP5-8.5 simulations was within a +/-0.2°C range of the target GWL (e.g. 1.3°C-1.7°C for the 1.5°C GWL).



The years chosen to represent each GWL differ slightly between SSP5-8.5 simulations. By centring timeslices on the target GWL, the ensemble average is very close to the target GWL in all cases and adequately representing that climate state. A

similar approach was taken for an "early stable" period (100-450 years post-emissions cessation) using NZ2035 to generate a 1.5°C GWL ensemble, NZ2045 to generate a 2°C ensemble, and NZ2060 to generate a 3°C ensemble. Corresponding "late stable" (650-1000 years post-emissions cessation) GWLs were extracted from the NZ2030 run to generate a 1.5°C ensemble, NZ2040 to generate a 2°C ensemble, and NZ2055 to generate a 3°C ensemble. This methodology accounts for the slow global warming in these simulations under net-zero emissions, but results may differ relative to using simulations where global

temperature trends are nearer zero for longer and the same simulations could be used to define the early and late stable GWL ensembles. The length of the millennial-scale simulations as well as having 40 available SSP5-8.5 simulations results in large ensembles being generated (Table 1). The method used to extract the GWLs is a timeslicing approach (Schleussner et al., 2016; King et al., 2017). The use of decades rather than longer averaging windows reduces overlap between the 1.5°C and 2°C GWLs in the transient simulations and is consistent with King et al., (2017).

The patterns of warming at the 1.5°C, 2°C and 3°C GWLs under transient (SSP5-8.5), in the early stable period, and in the late stable period were compared. This analysis was performed on boreal summer (June-August or JJA) and boreal winter (December-February or DJF) separately as previous work has identified differences in warming patterns by season (King et al., 2020, 2021b). Patterns of precipitation change at these GWLs were similarly analysed with percentage differences in seasonal precipitation between transient and the early and late stable period GWLs computed. Significance of differences was

estimated using a Kolmogorov-Smirnov test and a threshold p-value of 0.05. The large sample sizes and low autocorrelation between a given season from one year to the next means that the effective degrees of freedom values remain high (Wilks, 2011).

Following this, areas of significant reversal in precipitation trends were examined. This was done by identifying locations where the precipitation distribution is significantly different between the late stable period and SSP5-8.5 for a given GWL.

The direction (or sign) of this difference was then compared to the sign of difference between the precipitation distributions during 1850-1900 baseline and SSP5-8.5 runs at that GWL. Locations with opposing signs were classified as regions with a significant reversal in precipitation trend.

**Table 1. Number of years in each ensemble of global warming level by category.**

| GWL | Number of model years | | |
|---|---|---|---|
| | Transient | Early Stable | Late Stable |
| 3°C | 654 | 321 | 297 |
| 2°C | 691 | 361 | 341 |
| 1.5°C | 713 | 351 | 332 |



Differences between the reference GWLs in the Paris Agreement framework were also calculated. Seasonal-average temperature and precipitation differences between the 1.5°C and 2°C ensembles in SSP5-8.5 and in the early stable and late stable periods were computed. Difference maps were compared between the transient and stabilising climates.

### 2.3.3 Changes in extremes and variability

In addition to looking at mean-state changes as a function of time and GWL, changes in extremes and variability were also
examined to demonstrate the utility of these millennium-long simulations under near-stable global temperatures. Firstly, changes in monthly temperature extremes were examined by identifying the hottest calendar month at each gridcell in the 1850-1900 baseline period. As we used the 40-member ensemble of historical simulations there are 2040 monthly-mean temperature values for the hottest calendar month at each gridcell, so the standard deviation was computed and is well constrained with a large sample of data. A temperature threshold of the mean plus two times the standard deviation was used
to compute the frequency of temperature extremes in the early stable and late stable 1.5°C and 2°C GWLs. The difference in frequency of extremes between GWLs and between early and late stable periods was compared. This method for examining extremes changes follows that of Cassidy et al., (2023) and is similar to other previous studies (e.g. Thompson et al., 2022).

Finally, a basic analysis of ENSO frequency and variability changes was conducted. Previous work has suggested that ENSO variability is projected to increase under continued global warming in transient simulations (e.g. Cai et al., 2014, 2015), but a
study on longer-term ENSO behaviour under slowed global warming in fixed greenhouse gas concentrations simulations found a decrease in ENSO amplitude (Callahan et al., 2021). Here, the Niño-3.4 region sea surface temperatures (SSTs) were averaged over the July-June period in the transient (historical and SSP5-8.5 concatenated) and the stabilised simulations. The Niño-3.4 SST was detrended using a centred 15-year moving window in all transient and stabilised timeseries. The standard deviation of the detrended series was calculated in 100-year blocks which were compiled into four groupings: 20[th] century
(based on historical simulations), 21[st] century (based on historical and SSP5-8.5 simulations), NZ1-500 (based on the first five 100-year blocks in the stabilised simulations) and NZ501-1000 (based on the last five 100-year blocks in the stabilised simulations). The standard deviation was used to examine changes in ENSO amplitude. Changes in the frequency of El Niño and La Niña events were examined by counting events above +0.5°C and below -0.5°C, respectively in the same 100-year blocks. Kolmogorov-Smirnov test statistics were used to identify significant differences between distributions using a p-value
of 0.05.

### 3 Results

### 3.1 Time-varying projections





**Figure 3. Timeseries of a) decadal-average Land-Ocean temperature difference anomalies, b) decadal-average global mean precipitation anomalies, c) annual-average global mean sea surface temperature, d) annual-average global mean ocean temperature, e) decadal-average Arctic sea ice extent in March, f) decadal-average Arctic sea ice extent in September, g) decadal-average Antarctic sea ice extent in March, and h) decadal-average Antarctic sea ice extent in September. Values are shown in the ACCESS-ESM1-5 historical and SSP5-8.5 simulations (black) and the net-zero greenhouse gas emissions simulations (grading from yellow to red). Anomalies in a) and b) are computed from an 1850-1900 baseline.**





The seven 1000-year long simulations exhibit very slow changes in global-mean temperature such that they are suitable for use in examining the effects of climate stabilisation and differences with transient warming (Figure 1d). Beyond GMST, other changes are apparent. The global-average land-ocean temperature difference increases strongly under SSP5-8.5, but this trend slowly reverses in all of the net-zero emissions simulations (Figure 3a). Note that even after 1000 years under net-zero emissions, an increase of land-ocean temperature contrast relative to pre-industrial levels remains (Joshi et al., 2008).


As the planet as a whole and the global ocean surface on average warm very slowly under net-zero emissions, continued increases in global-average precipitation are simulated in all seven model runs (Figure 3b). The global-average precipitation rate is both a function of global warming and time (Andrews and Forster, 2010; Mitchell et al., 2016), and is illustrative of the continuing global-scale changes projected under net-zero emissions and global-average temperature stabilisation. Global-

average SSTs are projected to continue increasing (Figure 3c), consistent with the continued slow global-average warming and reducing land-ocean temperature contrast. The deeper ocean is projected to warm at a fast rate even centuries after emissions cessation causing the global-average vertically-averaged ocean temperature to increase (Figure 3d). The average temperature throughout the entire ocean increases to the extent that even under the lowest global warming simulation broadly aligning with the Paris Agreement, the average ocean temperature exceeds that of SSP5-8.5 at 2100 beyond 320 years after net-zero

emissions is achieved. The difference in average global ocean temperatures grows over time between the simulations highlighting the long-term effects of delay in achieving net-zero emissions. This is expected due to the timescales involved in ocean overturning circulation (Armour et al., 2016).

Arctic sea ice extent at the peak and trough of the seasonal cycle exhibits rapid reductions in the transient SSP5-8.5 simulations. Under net-zero emissions, Arctic sea ice change is relatively small with little consistent sign of deterioration or recovery

(Figure 3e,f). While there is little change in sea ice extent over the full length of the simulations, all simulations show apparent substantial trends of different signs over multi-decadal timescales in Arctic sea ice extent in March and September (Figure 3e,f). This is indicative of the significant internal variability and memory in sea ice extent that can give rise to the appearance of significant trends over shorter simulations, although there is evidence that ESMs generally overestimate persistence in Arctic sea ice extent anomalies (Giesse et al., 2021).

Antarctic sea ice extent also decreases under transient global warming, but unlike in the Arctic, also continues to decline through the net-zero simulations as well (Figure 3g,h). In September, around the peak of the seasonal cycle, Antarctic sea ice extent decreases faster under the simulations where emissions cessation is achieved later in the 21$^{st}$ century. While many CMIP6 models suffer from significant model biases in sea ice properties, ACCESS-ESM-1.5 performs well in capturing the mean and variability in Antarctic sea ice coverage over the historical period (Roach et al., 2020). Given reasonable model

performance, the incidence of low ice, or "ice-free" events was examined in the Arctic and Antarctic regions using the one million square kilometre threshold applied in previous work (Screen and Williamson, 2017; Kim et al., 2023). The likelihood





of sea ice-free Septembers in the Arctic is predominantly a function of global warming level with little change through the net-zero simulations (Figure 4a). In contrast, the occurrence of sea ice-free March events in Antarctica is a function of both global warming and time (Figure 4b). Under net-zero emissions, there is an increasing chance of sea ice free events in the Antarctic

to the point that there are more sea ice free events in the last century of NZ2030 than the first century of NZ2060, despite GMST being more than 1°C higher in the latter.

### a) Arctic sea ice free Septembers

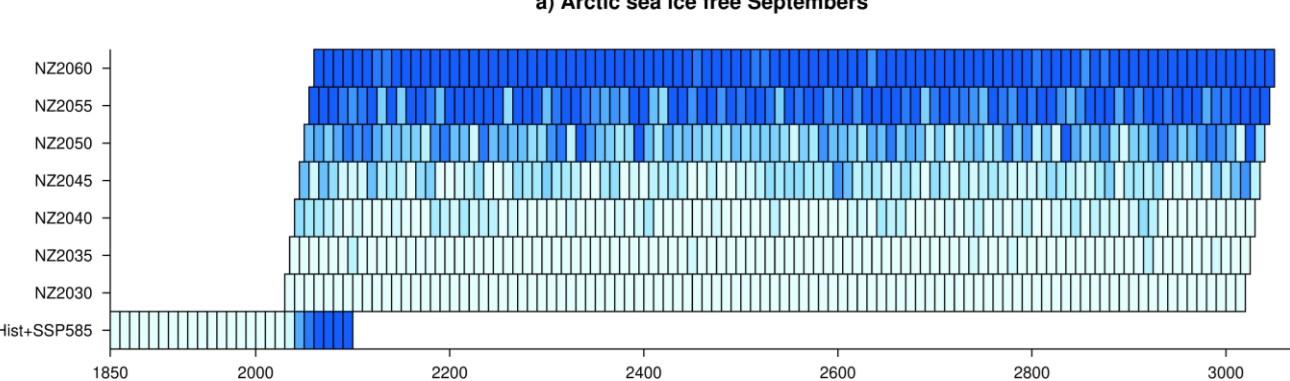

### b) Antarctic sea ice free Marchs

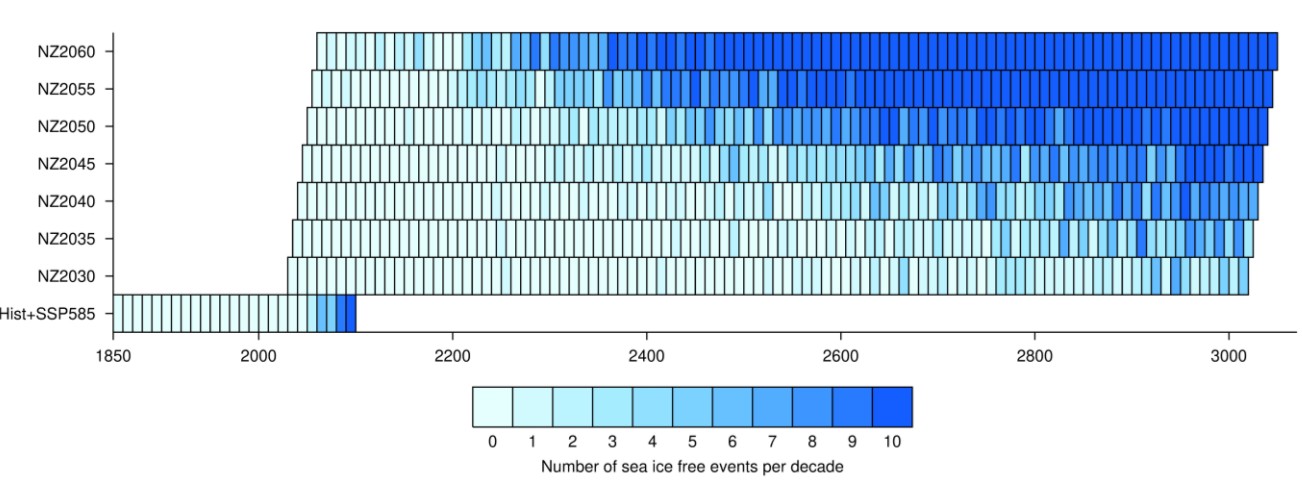

**Figure 4. Timeseries of number of ice-free months per decade in a) the Arctic in September and b) Antarctica in March. Timeseries are shown for the first concatenated historical and SSP5-8.5 simulation and each net-zero simulation. Ice-free months are defined**
**as months with average sea ice extent under one million square kilometres.**

Timeseries analysis of global indices, including land-ocean temperature difference and sea ice extent, suggests that there are complex changes occurring through the net-zero simulations despite minimal change in GMST. To explore this further we examine local-scale temperature changes. Under transient global warming, a clear pattern of faster warming over land is

identified, as well as strong Arctic amplification, whereas warming in the Southern Ocean is very slow (Figure 5a). As the



global temperature stabilises under net-zero emissions, this pattern evolves with land cooling relative to the global-average and the Southern Ocean warming (Figure 5b-e). This is consistent with Figure 3a, but the near-global nature of land cooling relative to GMST is remarkable. The northern Indian Subcontinent is an exception with warming relative to the global-mean in the net-zero simulations compared to the transient case. This is likely related to the reduced atmospheric anthropogenic

aerosol forcing in this region in the net-zero emissions runs relative to SSP5-8.5, but also perhaps a response to Southern Ocean warming (Oh et al., 2022). The continued Southern Ocean warming helps to explain the continued decrease in Antarctic sea ice extent under net-zero emissions (Figure 3g,h, 4).

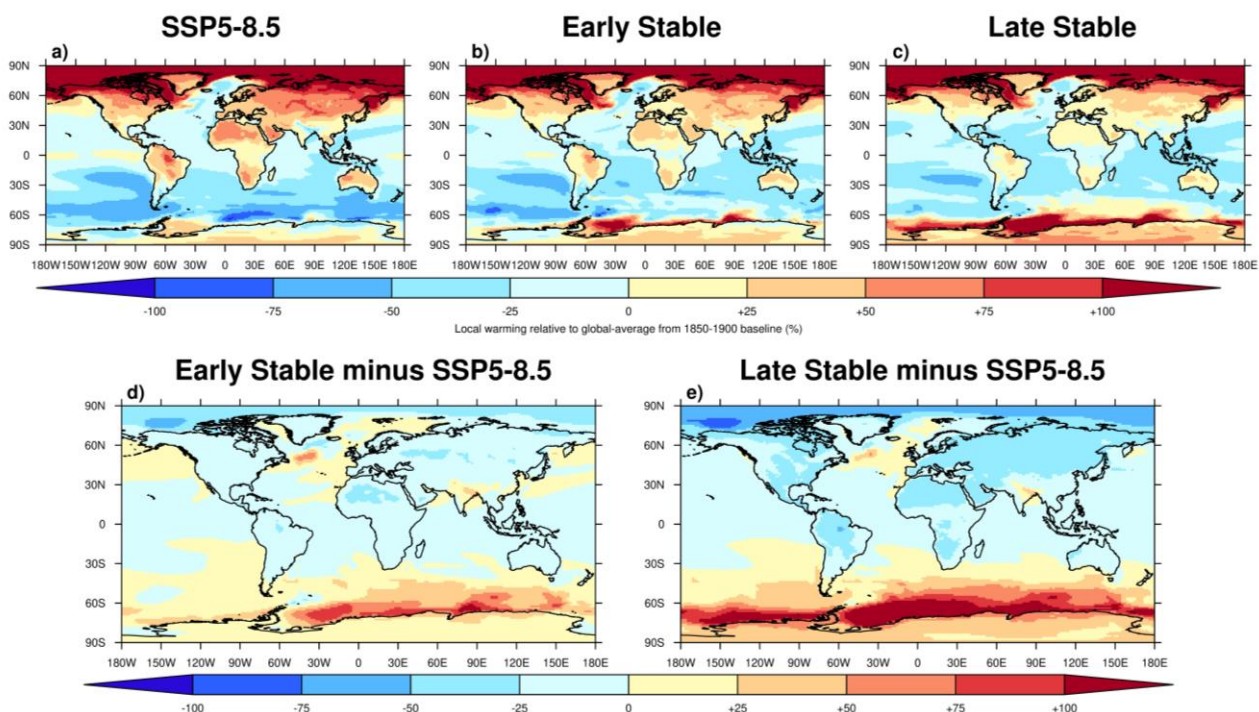

**Figure 5. Maps of annual local temperature change relative to the global-average for a) SSP5-8.5 in 2030-2069, b) all net-zero**
**simulations 200-399 years after emissions cessation, and c) all net-zero simulations 800-999 years after emissions cessation. Maps of the difference between the patterns of warming in d) all net-zero simulations 200-399 years after emissions cessation relative to SSP5-8.5 in 2030-2069, and e) all net-zero simulations 800-999 years after emissions cessation relative to SSP5-8.5 in 2030-2069.**

While land is projected to cool and ocean is projected to warm relative to GMST, there is also slow global-average warming
in these net-zero simulations. This results in projected absolute warming over most of the planet, both land and ocean (Figure 6a). The warming is weak over most land areas but exceeds 2°C over much of the sea ice region in the Southern Ocean between the early and late periods after emissions cessation. We have also already seen that the effect of a delay in achieving net-zero emissions continues through the millennium-length simulations and even grows in some variables. It is remarkable that in the last two centuries of the simulations there are still higher temperatures locally across the planet when there is only a five-year
delay in reaching net-zero emissions (Figure 6b). The difference is particularly large in the planet's sea ice regions,





emphasising the role of ice-albedo feedbacks in amplifying small temperature anomalies. This result suggests there are major long-term costs associated with even a short delay in achieving net-zero emissions.

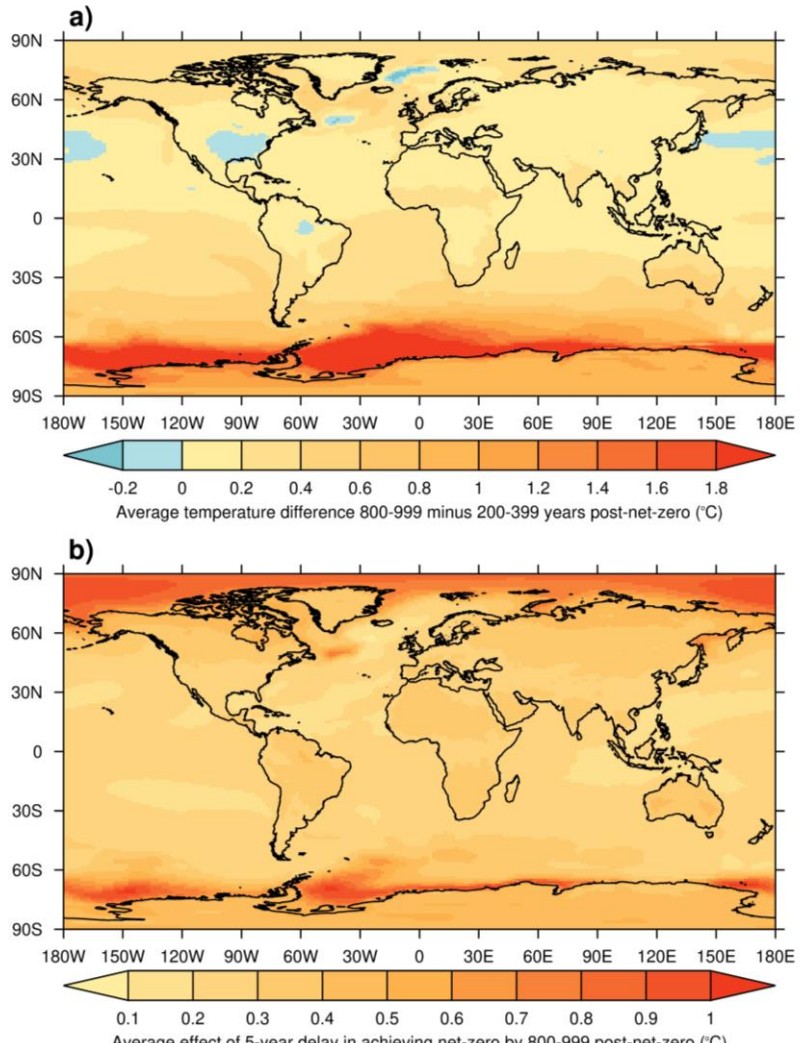

**Figure 6. a) Map of annual-average temperature difference between years 800-999 and 200-399 averaged across all net-zero simulations. b) Map of average difference between net-zero simulations where emissions cessation is delayed five years (i.e. NZ2035-NZ2030, NZ2040-NZ2035, NZ2045-NZ2040, NZ2050-NZ2045, NZ2055-NZ2050 and NZ2060-NZ2055) in years 800-999 post emissions cessation.**

Different areas of the world experience markedly varied climate trajectories beyond net-zero emissions. To investigate this in more detail, the timing of peak temperatures relative to the point of emissions cessation was plotted (Figure 7a). For a few areas, primarily in Africa, we find that temperatures peak before the point of net-zero emissions. Many Northern Hemisphere land regions experience peak warming within a couple of centuries of emissions cessation, but in the Southern Hemisphere





land regions there is a substantial delay to local temperatures peaking. Unsurprisingly, given the previous results, the Southern Ocean peak warming is consistently in the last couple of centuries of the simulations and is likely limited by the runs ending 1000 years after emissions cessation. This is consistent with the slow changes found using other long simulations in previous analyses (Armour et al., 2016; Grose and King, 2023). For individual city locations (Figure 4b-e) we see relatively small temperature changes projected after net-zero relative to the preceding period, but for the example of Melbourne (Figure 4e) there is still as much as 1°C of post-net-zero warming in the NZ2060 simulation up to a millennium after emissions cessation.

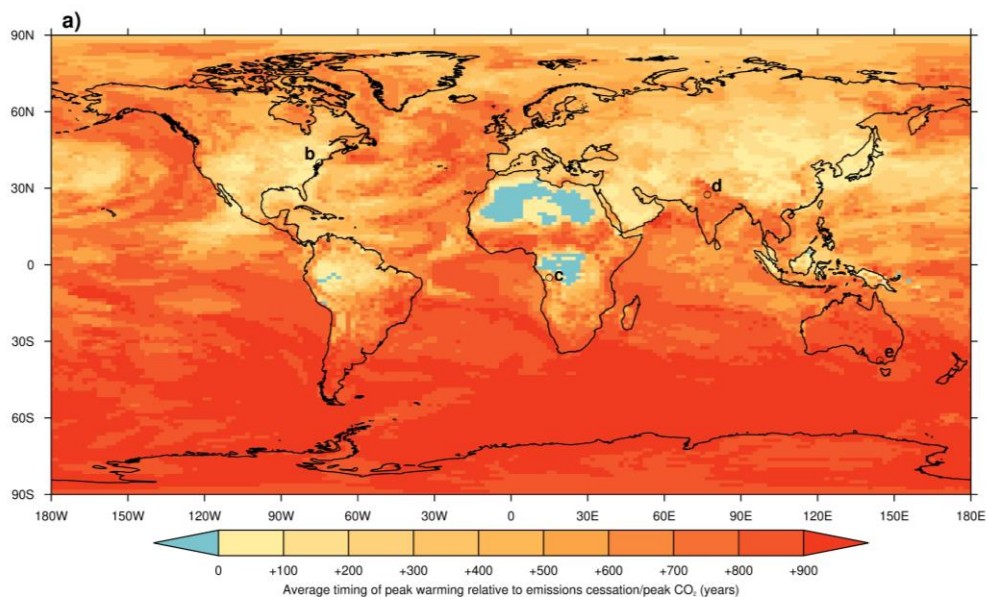

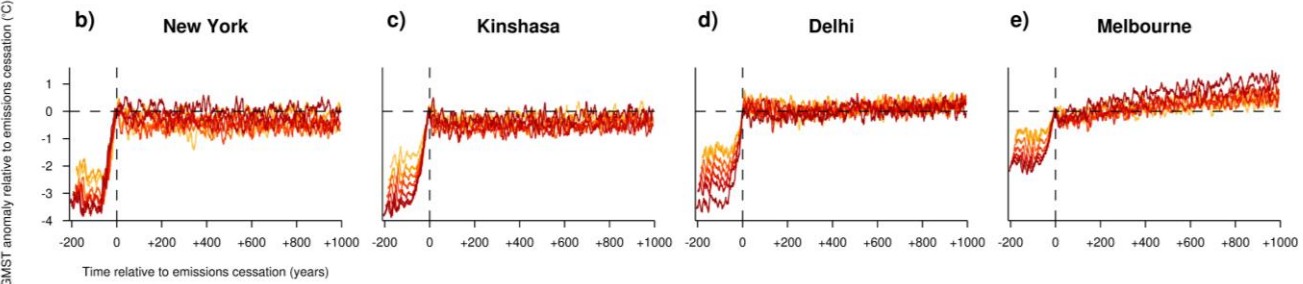

**Figure 7. a) Map of average timing of local peak 11-year smoothed average annual temperatures relative to the point of emissions cessation. b-e) Timeseries of 11-year smoothed temperatures relative to the point of emissions cessation in each net-zero simulation from NZ2030 (yellow) to NZ2060 (red) in the gridcells over b) New York City, c) Kinshasa, d) Delhi, e) Melbourne. Locations of these cities are shown on a). Note the gridcell for Melbourne is north of the city so it is a land-only gridcell.**

## 3.2. Global warming level projections

The length and slow-evolving nature of these net-zero emissions simulations means they are suitable for investigating the implications of different GWLs and dependence on rate of global warming. Prior to this study, such analysis has been





challenging and relied on statistical techniques with assumptions (King et al., 2021b, 2020). First, the pattern of seasonal-average temperature changes between the rapid-warming SSP5-8.5 ensemble and the early and late parts of the net-zero

simulations (see Figure 2) was examined.

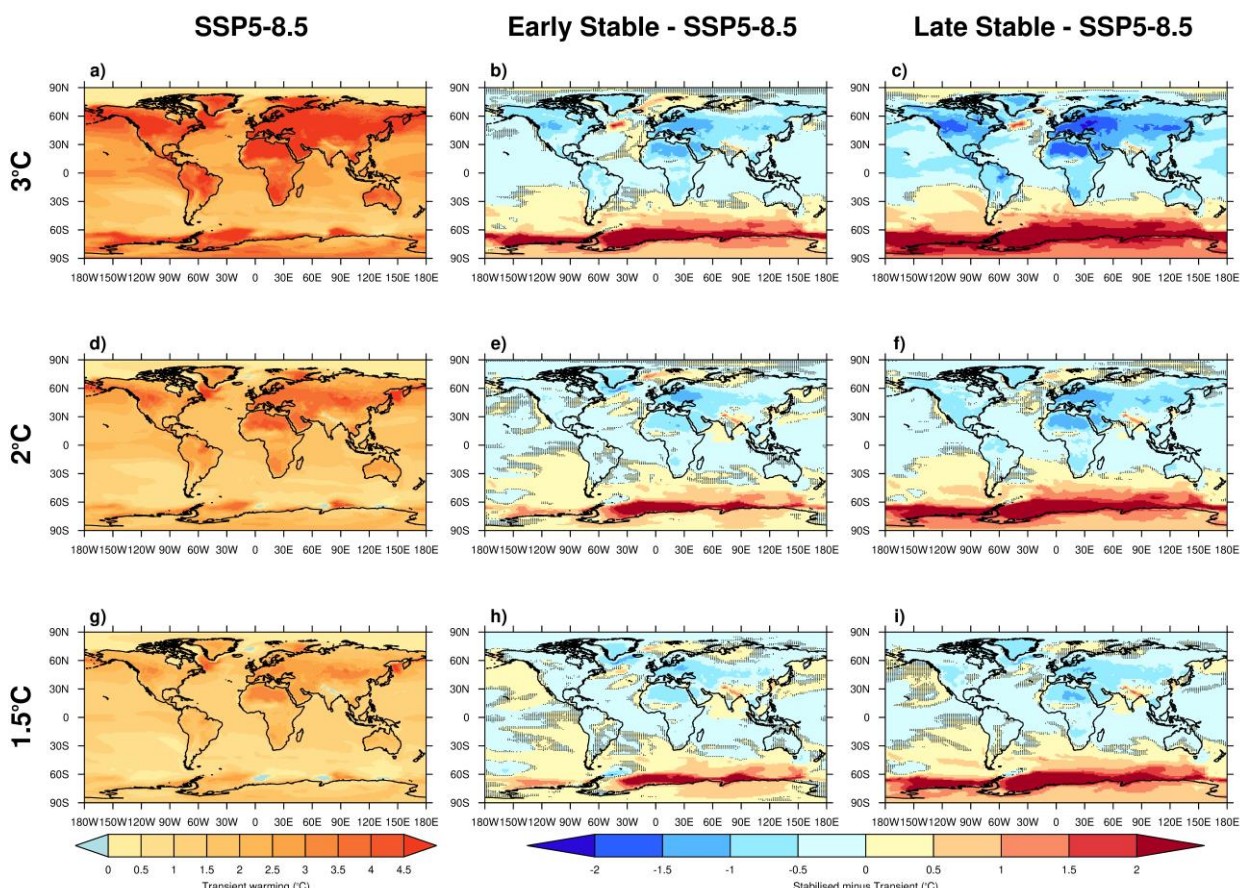

**Figure 8. Warming in boreal summer (JJA) average temperatures at a) 3°C, d) 2°C and g) 1.5°C global warming levels in SSP5-8.5 simulations relative to the 1850-1900 baseline. The difference in JJA average temperatures between global warming levels extracted between 100 and 450 years after emissions cessation and SSP5-8.5 at b) 3°C, e) 2°C and h) 1.5°C global warming levels. The difference**

**in JJA average temperatures between global warming levels extracted between 650 and 1000 years after emissions cessation and SSP5-8.5 at c) 3°C, f) 2°C and i) 1.5°C global warming levels. b), c), e), f), h), i) Stippling shows where distributions are not significantly different at the 5% level using a Kolmogorov-Smirnov test.**

There are differences in the transient warming pattern between boreal summer (JJA; Figure 8) and boreal winter (DJF; Figure

9) with Arctic amplification absent in JJA (Lu and Cai, 2009) and a tendency for greater land-surface warming in the

summertime hemisphere. Generally, in SSP5-8.5, there is scalability in local temperature changes with global changes such

that the warming pattern at 3°C global warming is approximately double the warming pattern at the 1.5°C GWL (Seneviratne

et al., 2016). There are differences in the warming pattern between the early post-net-zero period and SSP5-8.5 and these grow



for the later post-net-zero period. The differences are also seasonally dependent (King et al., 2020) and larger for higher GWLs.
In DJF, the difference between sampling the 3°C GWL from the later stable period relative to the SSP5-8.5 simulations exceed
more than 2°C across much of the Southern Ocean and the Arctic (Figure 9c). In JJA, there are many mid-latitude land regions
for which sampling a 3°C GWL from a transient or stabilised climate results in differences of more than 1.5°C in local seasonal-
average temperatures. When GWLs are sampled from stabilised runs as opposed to transient runs, temperatures in the Southern
Ocean are substantially higher with greater than 2°C higher temperatures across much of the Southern Ocean regardless of
season. This is consistent with the ongoing warming and reduction in sea ice extent identified previously.

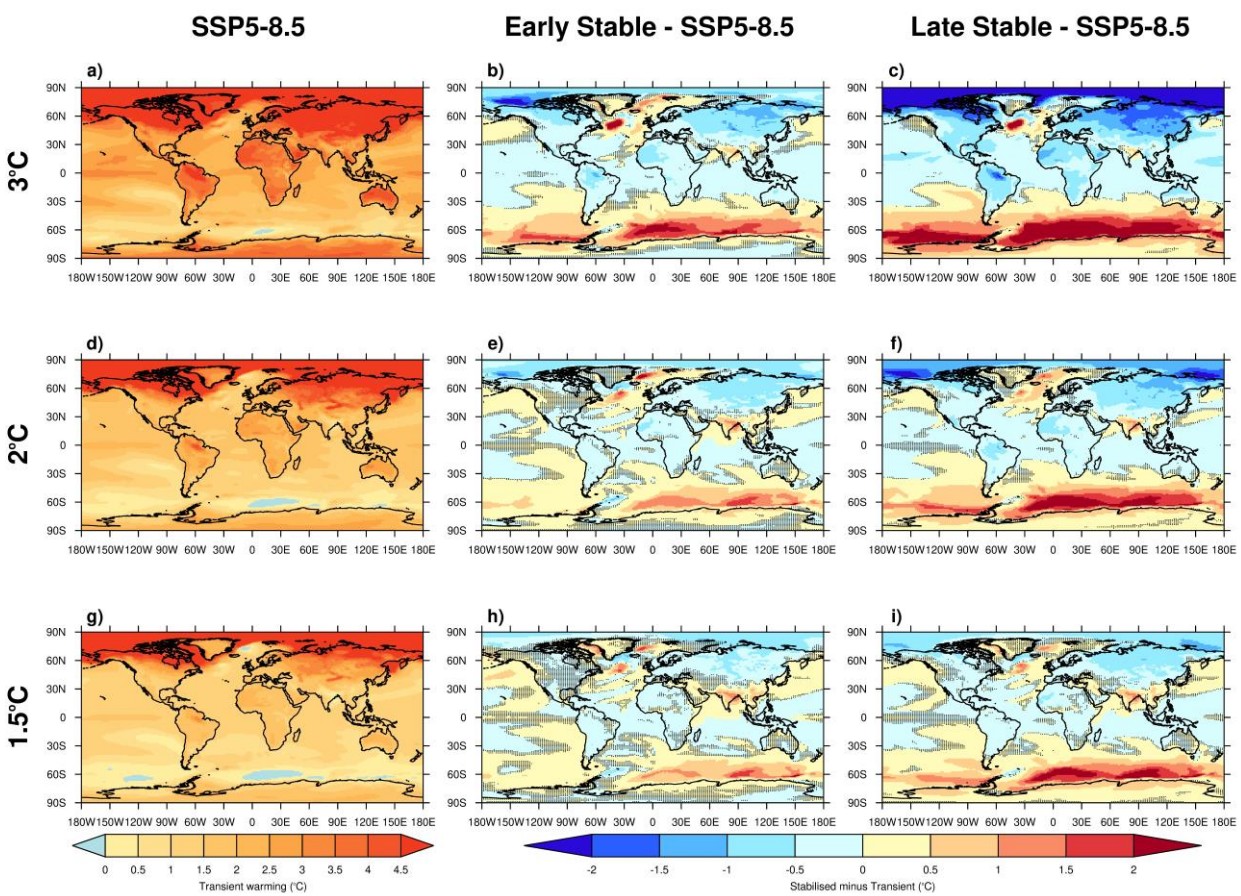

**Figure 9. As Figure 8 but for boreal winter (DJF).**

Previous work has identified precipitation changes under climate stabilisation (Ceppi et al., 2018; Grose and King, 2023;
Sniderman et al., 2019). Using these ACCESS-ESM-1.5 millennium-length simulations we also identify significant differences
in precipitation change patterns between transient and stabilising GWL projections (Figure 10, 11). In some cases this is
indicative of transient trends intensifying under stabilisation (such as the projection of increased austral winter precipitation





over Antarctica; Figure 10) while in other cases there is projected reversal, particularly over sub-tropical ocean regions projected to dry under continuing greenhouse gas emissions but which are significantly wetter under prolonged net-zero
emissions at the same GWLs (Figures 10, 11).

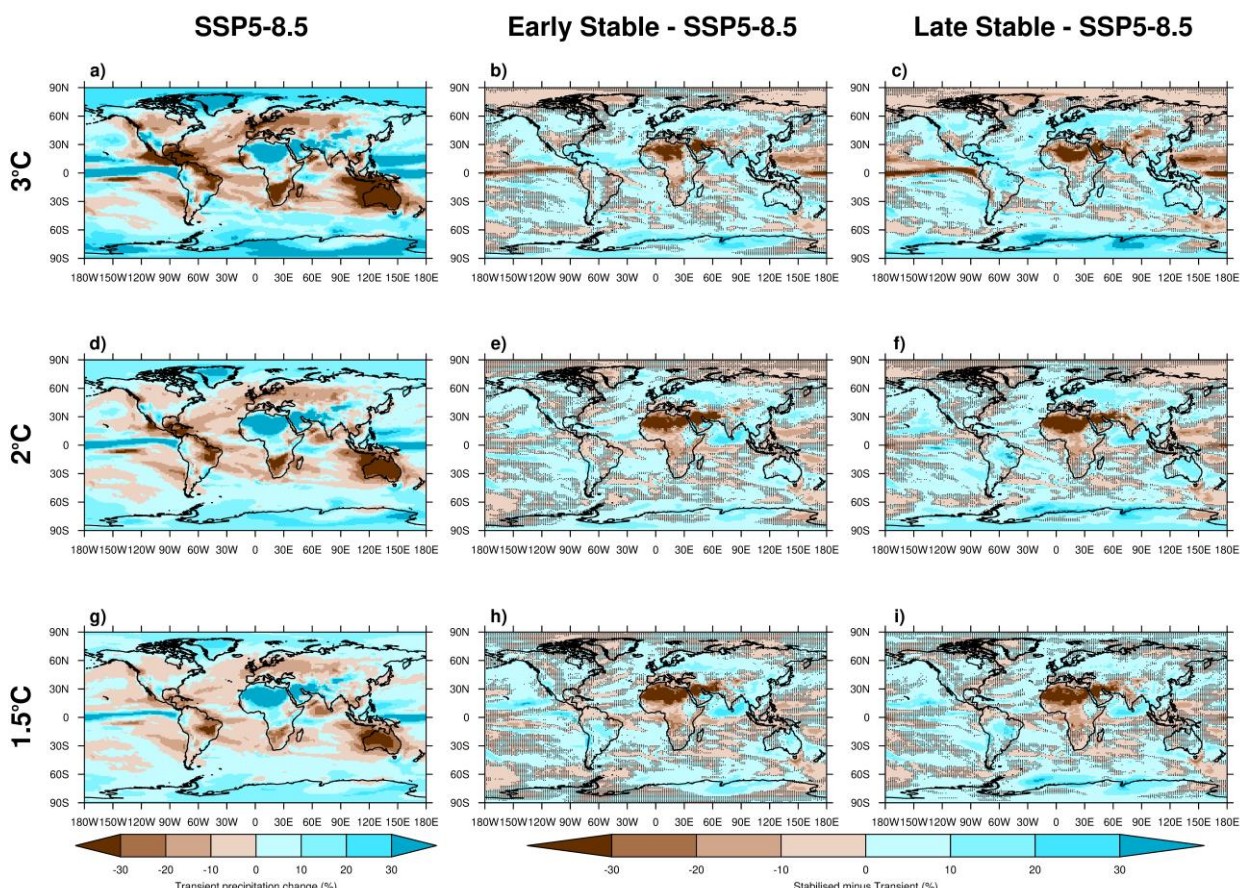

**Figure 10. Change in boreal summer (JJA) average precipitation at a) 3°C, d) 2°C and g) 1.5°C global warming levels in SSP5-8.5 simulations relative to the 1850-1900 baseline. The difference in JJA average precipitation between global warming levels extracted between 100 and 450 years after emissions cessation and SSP5-8.5 at b) 3°C, e) 2°C and h) 1.5°C global warming levels. The difference in JJA average precipitation between global warming levels extracted between 650 and 1000 years after emissions cessation and SSP5-8.5 at c) 3°C, f) 2°C and i) 1.5°C global warming levels. b), c), e), f), h), i) Stippling shows where distributions are not significantly different at the 5% level using a Kolmogorov-Smirnov test.**



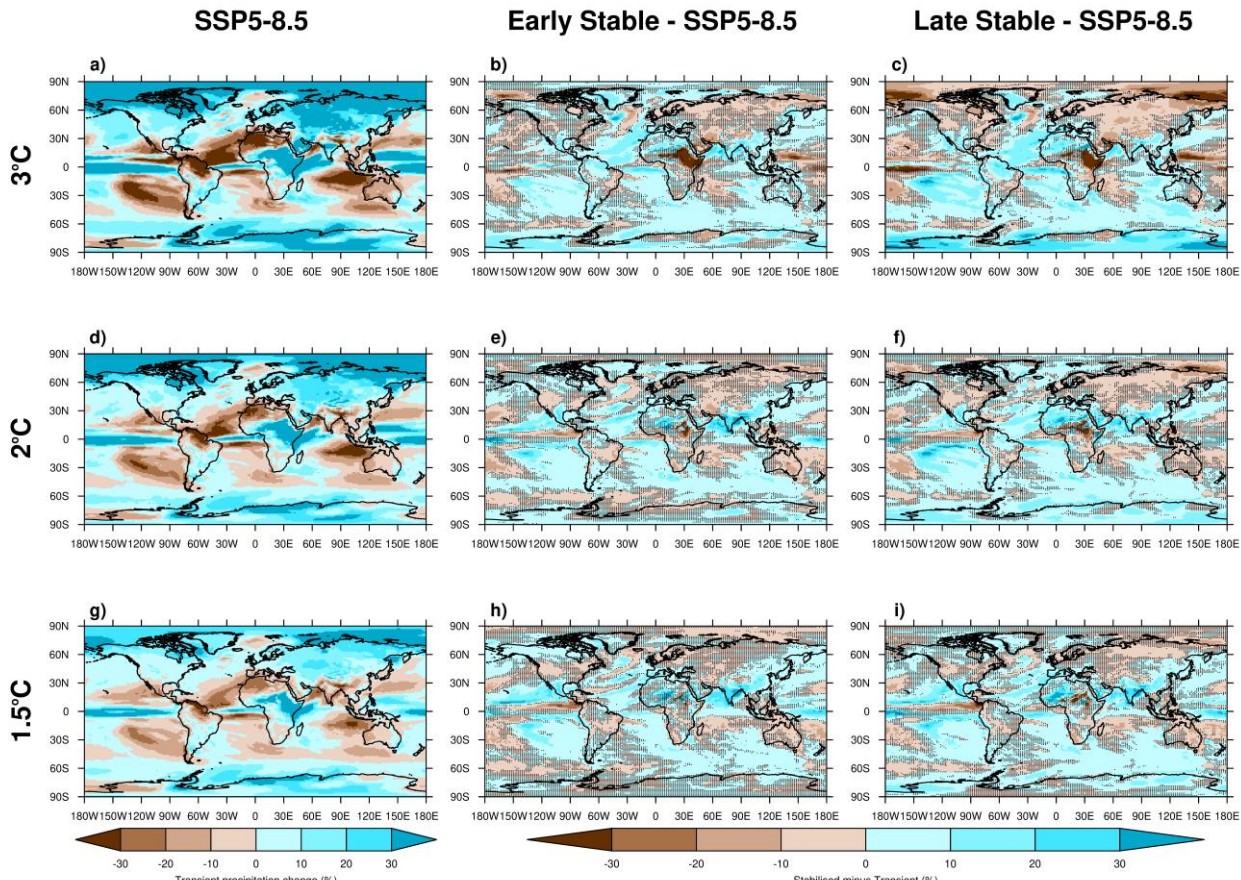


**Figure 11. As Figure 10 but for boreal winter (DJF).**

Areas of significant precipitation trend reversal are shown in Figure 12 and suggest very large areas of the world may exhibit some return towards pre-industrial levels of seasonal-average precipitation. This includes many areas of the world which have

been extensively studied due to concern about drying under continued climate change, including the Mediterranean region (Giorgi and Lionello, 2008; Gudmundsson and Seneviratne, 2016) and much of Australia, where stabilisation at a high GWL is projected to result in significant June-August precipitation increase (Figure 12a). In contrast, a commonly identified projection of rainfall increase over the Sahara in boreal summer (Pausata et al., 2020) reverses significantly as the climate stabilises (Figure 10, 12). There is a general pattern of larger trends in precipitation at higher GWLs under transient climate

change (Figures 10,11), and also greater area of significant reversals in trends at higher GWLs (Figure 12).





**Figure 12. Locations of reversal in precipitation trends under transient and stabilised climate changes for a), b) 3°C, c), d) 2°C and e, f) 1.5°C global warming levels in JJA and DJF respectively. Locations in blue are drier under SSP5-8.5 but are significantly wetter (p<0.05) by 650-1000 years after emissions cessation at the same global warming level. Locations in brown are wetter under SSP5-8.5 but are significantly drier (p<0.05) by 650-1000 years after emissions cessation at the same global warming level. Percentage of the global surface with significant precipitation reversal are shown for significant increase under stabilisation relative to SSP5-8.5 (blue) and significant decrease under stabilisation relative to SSP5-8.5 (brown).**

Much of the interest in global warming level-based climate projections was initially focussed on identifying and understanding differences in climate between the GWLs referred to in the Paris Agreement: 1.5°C and 2°C above pre-industrial levels. Thus, here we examine whether differences between these GWLs vary depending on using transient or stabilised simulations. The



ACCESS-ESM-1.5 model simulates a pattern of transient warming and precipitation changes between 1.5°C and 2°C GWLs
(Figure 13) that is consistent with previous multi-model ensemble-based findings (King et al., 2017; Masson-Delmotte et al.,
440    2018).

**Figure 13. Differences in a) JJA and d) DJF temperatures and g) JJA and j) DJF precipitation between 1.5°C and 2°C global warming levels in SSP5-8.5. Differences between 1.5°C and 2°C temperature and precipitation changes for GWLs drawn from 100-450 years after emissions cease and SSP5-8.5 in b), e), h) and k). Differences between 1.5°C and 2°C temperature and precipitation changes for GWLs drawn from 650-1000 years after emissions cease and SSP5-8.5 in c), f), i) and l). Hatching illustrates where corresponding absolute differences between stabilised and transient temperature or precipitation are smaller than absolute differences between 1.5°C and 2°C global warming levels in SSP5-8.5.**


As the climate stabilises the pattern of differences between 1.5°C and 2°C GWLs evolves. Over most land areas the difference
in seasonal-average temperatures between GWLs under transient warming is substantially greater than between corresponding
stabilised GWLs. Over parts of the Southern Ocean, the local differences in JJA and DJF average temperatures between



stabilised 1.5°C and 2°C GWLs are more than double the differences between transient 1.5°C and 2°C GWLs. These results

are similar to those identified by King et al., (2020). For seasonal-average precipitation, differences between 1.5°C and 2°C

GWLs are more subtle and for many parts of the world the effect of sampling from stabilised or transient GWLs is greater than

the effect of a 0.5°C difference in global temperatures. These results highlight the importance of framing projections based on

GWLs very clearly to reduce the likelihood of misinterpretation.

**3.3. Climate Extremes**

We have demonstrated the utility of these millennium-long ACCESS-ESM1.5 simulations for understanding mean climate

changes and climate state dependence on the rate of global warming. Here, we also briefly examine two further applications

of these simulations for analysis of changes in climate extremes and climate variability.

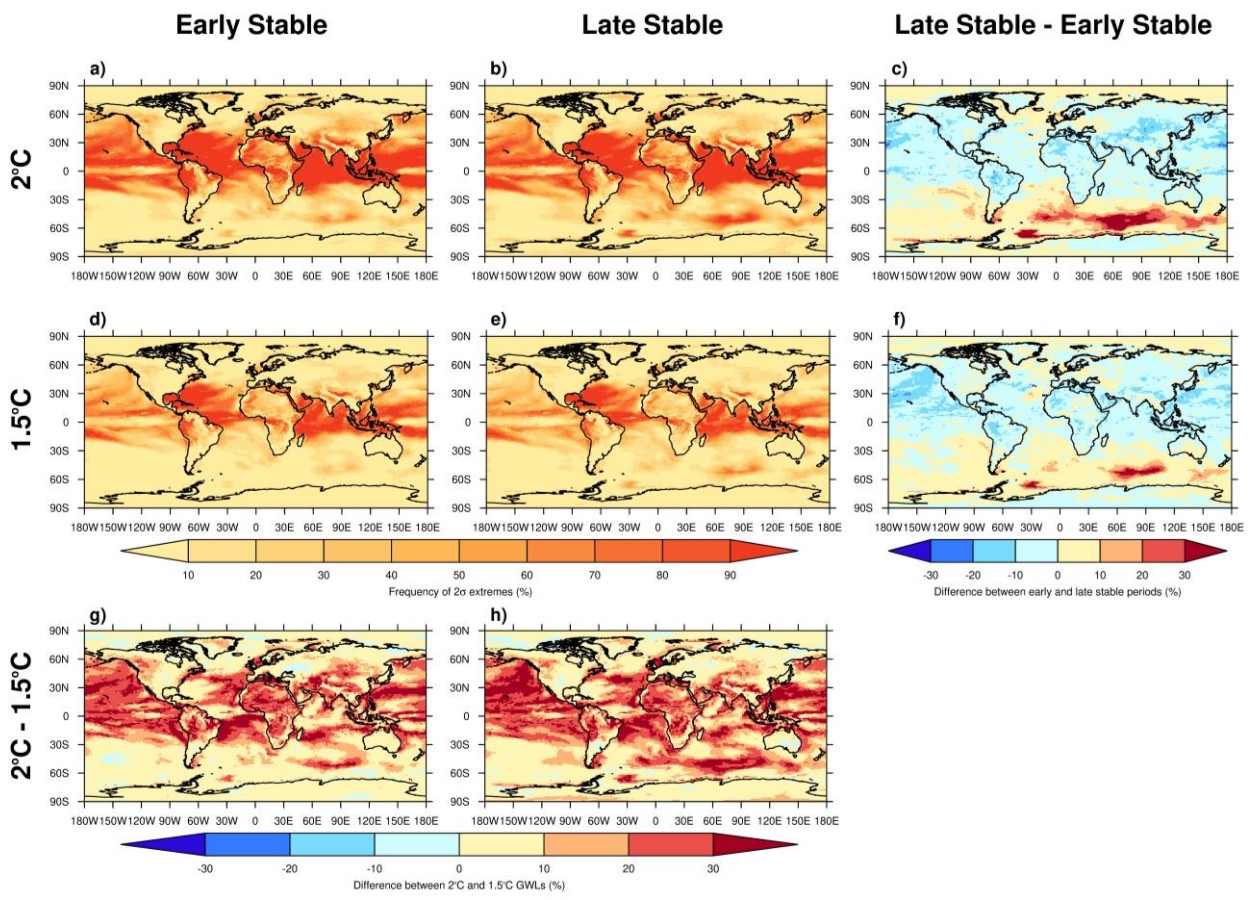

**Figure 14. Frequency of exceeding 2σ above the 1850-1900 average temperature in the climatological hottest month of the year at each location at a), b) 2°C GWL drawn from 100-450 years and 650-1000 years after emissions cease respectively. e), f) is the same but for the 1.5°C GWL. c), f) Difference in frequency of extreme hot months between GWLs drawn from 100-450 years and 650-1000 years after emissions cease for 2°C and 1.5°C respectively. G), h) Differences in frequency of extreme hot months between 2°C and 1.5°C GWLs drawn from 100-450 years and 650-1000 years after emissions cease, respectively.**






An analysis of the frequency of hot months at different GWLs and times in the stabilised runs is used to demonstrate the types of studies that are possible using these simulations. The hottest calendar month of the year at each location in the ACCESS-ESM-1.5 historical 1850-1900 period was identified and the variability in that month's temperatures was used to define a threshold of the mean plus two standard deviations for investigating extremes exceedance. The frequency of extremes above

this threshold is unsurprisingly higher in the 2°C GWL samples than in the 1.5°C GWL samples, regardless of whether these are drawn from earlier or later on in the net-zero simulations (Figure 14). A higher frequency of extremes is projected in the tropics and low-latitude ocean areas where signal-to-noise ratios are greater (Hawkins and Sutton, 2012; Harrington et al., 2016; Mahlstein et al., 2011; Hawkins et al., 2020). The low-latitude regions with high signal-to-noise are also where the greatest increase in frequency of temperature extremes is projected between the 1.5°C and 2°C GWLs (Figure 14g,h). Under

stabilisation, there is a marked reduction in the frequency of hot months over many low-to-mid-latitude regions, but an increase in frequency of hot months over the Southern Ocean (Figure 14c,f). Larger differences may be identifiable comparing transient and stabilised 1.5°C and 2°C GWLs rather than different times within the net-zero simulations, but there are challenges in robustly estimating extremes frequency and intensity from fast-warming simulations (King et al., 2020).

## 3.4. El Niño-Southern Oscillation

El Niño-Southern Oscillation (ENSO) is the most important mode of climate variability on interannual timescales and changes in ENSO could have far-reaching implications given its teleconnections to regional climates and impacts (Yeh et al., 2018; Lieber et al., 2022). Here, we examined how ENSO amplitude and frequency of El Niño and La Niña events compares between transient and stabilised climate states. ENSO amplitude, as measured by the Niño-3.4 SST standard deviation (Callahan et al., 2021), is projected to significantly increase under continued global warming (Figure 15a). In contrast, after emissions cessation

(in both the earlier and later 500-year blocks) the amplitude of ENSO is not significantly different from that simulated for the 20[th] century and significantly lower than in transient 21[st] century simulations. This is in broad alignment with previous literature. Cai et al., (2022) found an increase in ENSO amplitude under high and low emission transient simulations in CMIP6 while Callahan et al., (2021) identified decreased ENSO amplitude under climate stabilisation in a multi-model ensemble of simulations forced by fixed-$CO_2$ concentrations. There is also evidence of temporal variation in ENSO amplitude which differs

between models (Maher et al., 2023), so further analysis and study of model dependence would be useful.



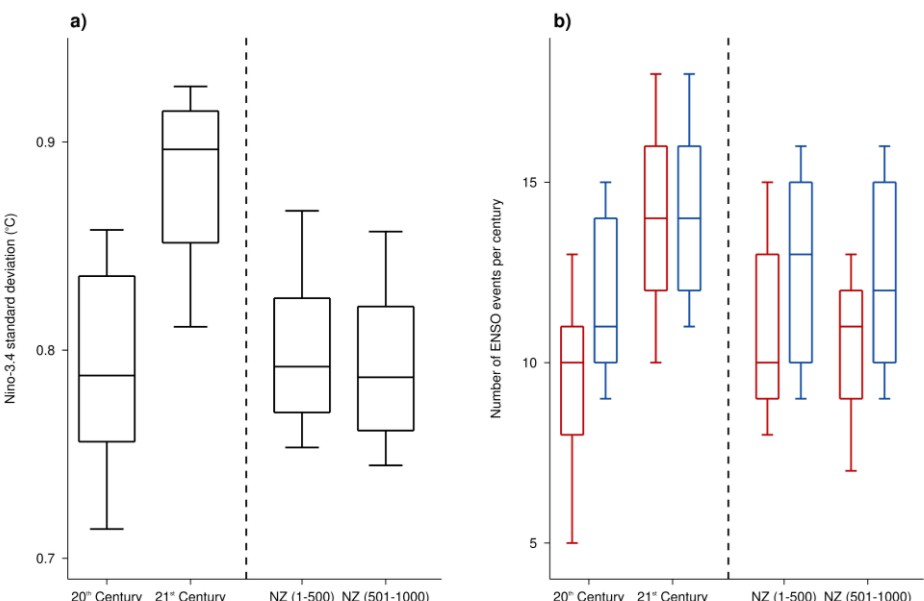

**Figure 15. a) Box plots of standard deviation in Niño-3.4 average temperatures in the 20th century and 21st centuries in fast-warming simulations, and in the first and second 500 years after emissions cessation. b) Box plots of El Niño (red) and La Niña (blue) frequency per century in these simulations.**


The frequency of El Niño events follows a similar trajectory with increased occurrence under a continued high greenhouse gas emissions scenario and reduced frequency after net-zero emissions is achieved (Figure 15b). On the other hand, La Niña events are not projected to increase in frequency significantly in the 21$^{st}$ century fast-warming simulations and would be expected to decrease slightly, albeit non-significantly, under net-zero emissions. These projections point to contrasting ENSO

characteristics between transient and stabilised warmer worlds with likely effects on regional climates beyond the tropical Pacific Ocean. Further work is needed to explore the mechanisms for ENSO changes under transient versus net-zero conditions.

**4. Summary and Conclusions**

A new set of 1000-year long ACCESS-ESM-1.5 simulations under net-zero carbon dioxide emissions has been run to help inform critical decision-making around the long-term implications of our current policy goals. We believe that these

simulations are unique at present with respect to length and low warming level stabilisation allowing for analysis of the climate states associated with the Paris Agreement GWLs. This framework complements existing modelling efforts, including ZECMIP (Jones et al., 2019) and LongRunMIP (Rugenstein et al., 2019), but these new simulations are integrated for longer, the parent simulation is a scenario-based climate, and there are a range of branching points so that the simulated climates span a range of warming levels on both short and long timescales.

Within this framework, net-zero carbon dioxide emissions is insufficient to prevent further global warming in ACCESS-ESM-1.5 as continued slow global warming beyond the point of emissions cessation is identified. Initial analysis of these ACCESS-



ESM-1.5 simulations suggests that the regional climate response to net-zero emissions would be diverse and evolve over subsequent centuries. This work, along with previous studies, highlights the limitations of GMST-based climate targets as different places and populations experience different climate changes under both continued global warming (Harrington et al., 2018) and in a post-net-zero climate state.

Various properties of the climate system evolve differently under sustained net-zero emissions. While there may be benefits of reduced land temperature means and extremes as well as reversal of some regional precipitation trends, ocean warming is projected to continue. The Southern Hemisphere high latitudes show particularly slow responses to net-zero emissions with long-term warming for many centuries projected beyond emissions cessation accompanied by continuing Antarctic sea ice decline. Extratropical locations in the Southern Hemisphere, such as Melbourne, are projected to warm for centuries after emissions cessation in stark contrast to other land areas. While this study does not explore sea level rise, it is already known that this will continue for many centuries under net-zero emissions pathways (Nauels et al., 2019). It is clear that the climate of a post-net-zero world will pose different regional risks across the world compared with the near-term transient warming climate and that humanity must prepare so that these risks may be mitigated. This study also suggests that any delay to achieving net-zero emissions may have long-lasting consequences and make achievement of low GWLs more challenging without substantial net-negative emissions. This adds to the large body of evidence (IPCC, 2021) showing benefits to earlier emissions reductions and achievement of net-zero carbon dioxide emissions.

The results shown here are based on the ACCESS-ESM-1.5 model only. While the ACCESS-ESM-1.5 model performs reasonably well in general (Ziehn et al., 2020), this is a single-model analysis and the results should be interpreted in this context. The capability of ESMs to simulate slow-evolving changes in the Earth system or potential tipping points has been debated (Armstrong McKay et al., 2022) and this is a relevant concern given recent rapid changes observed in Antarctic sea ice (Purich and Doddridge, 2023). The results presented in this study use one of our best available modelling tools to understand future climates under net-zero emissions, but improved understanding of slow climate processes and the potential for sudden-onset changes is needed.

The hope with this model framework is that other groups might consider running similar simulations (King et al., 2021a). As discussed previously, other experiments are being run with different protocols but may provide complementary data for analysis. The framework employed here is highly idealised, so future work running simulations under more plausible scenarios and including prolonged net-negative emissions would be beneficial.

This model ensemble may be used to answer many critical questions about future climate changes projected under net-zero emissions. The long simulations with near-stable global temperatures and the pre-existing large ensemble of historical and SSP5-8.5 simulations support many potential analyses to improve understanding of climate change, variability and extremes under slow and fast-changing climate states. This study attempts to demonstrate the utility of these simulations for exploring





changes in extremes and in variability, but further work is needed to comprehensively understand climate changes beyond emissions cessation. We encourage analysis of these simulations to help improve understanding of changes in other forms of

climate variability, teleconnections, and extremes, ocean and cryosphere properties, and the carbon cycle.

**Appendices**

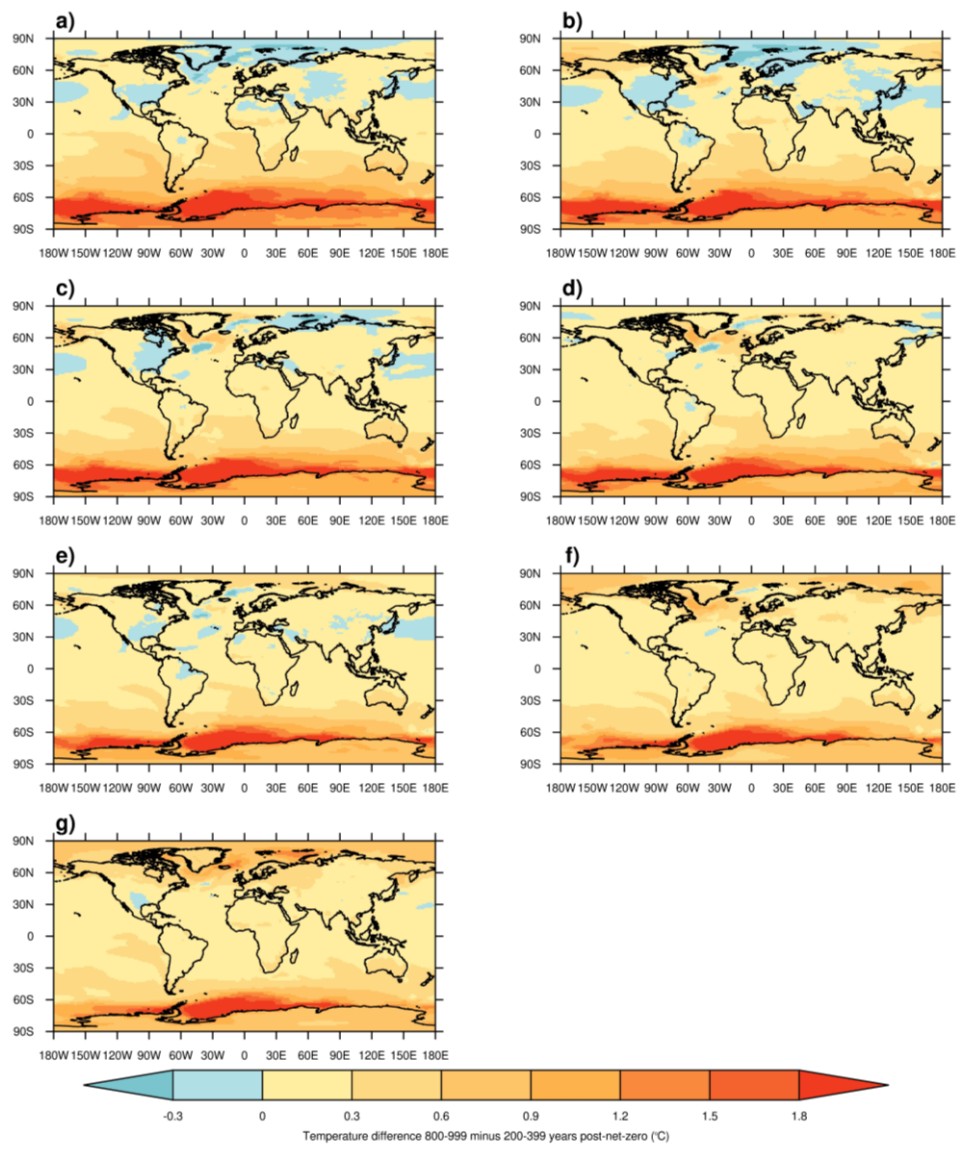

**Figure A1. Maps of annual-average temperature difference between years 800-999 and 200-399 in a) NZ2060, b) NZ2055, c) NZ2050, d) NZ2045, e) NZ2040, f) NZ2035 and g) NZ2030.**






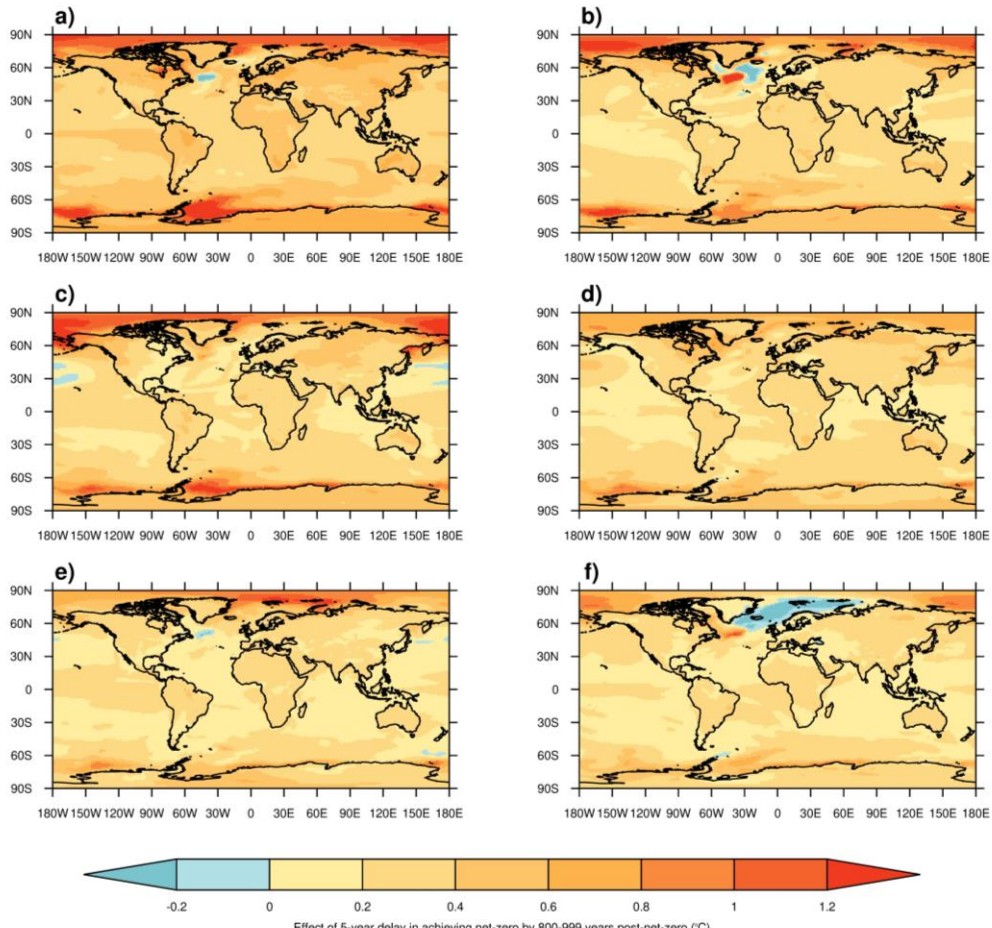

**Figure A2. Maps of annual-average temperature differences between net-zero simulations where emissions cessation is delayed five years in years 800-999 post emissions cessation for a) NZ2060-NZ2055, b) NZ2055-NZ2050, c) NZ2050-NZ2045, d) NZ2045-NZ2040, e) NZ2040-NZ2035, and f) NZ2035-NZ2030.**

### Data Availability

All model simulations are available on the Australian node of the Earth System Grid Federation. The historical simulations are available here: https://www.wdc-climate.de/ui/cmip6?input=CMIP6.CMIP.CSIRO.ACCESS-ESM1-5. The SSP5-8.5 simulations are available here: https://www.wdc-climate.de/ui/cmip6?input=CMIP6.ScenarioMIP.CSIRO.ACCESS-ESM1-5. Data from the stabilised runs described is being post-processed and will be made available once final revisions are made. All code used to analyse the data and generate the plots will be made available once final revisions are made.

### Author contributions



ADK and TZ designed the simulations. TZ performed the simulations. ADK led the writing of the manuscript and performed the analysis. All authors contributed to the interpretation of results and writing of the manuscript.

**Competing interests**

The authors declare no competing interests.

**Acknowledgements**

ADK, TZ, MC, JRB and MG receive funding from the Australian Government through the National Environmental Science Program. SP-K is funded through ARC grant number FT170100106. The simulations were performed using the Australian

National Computing Infrastructure (NCI).

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
