# Peer review of "Exploring climate stabilisation at different global warming levels in ACCESS-ESM-1.5"

_EGUsphere, 2023_

## Author Comment (AC1)

**General comments**

This manuscript analyzes a new set of climate change experiments utilizing the ACCESS-ESM-1.5 Earth system model to simulate climate evolution under net-zero emissions. Seven simulations spanning 1000 years were branched of the SSP5-8.5 simulation at 5-year intervals starting in 2030 to assess climate stabilization in line with the Paris Agreement and higher warming scenarios. Significant findings are provided for continued Southern Ocean warming, reversal of mid-latitude drying trends, and varying sea ice extent trajectories in the Arctic and Antarctic. The study also examines the El Niño-Southern Oscillation (ENSO), noting reduced amplitude and frequency of events under climate stabilization. The results show that regional changes persist for centuries post-emission cessation, indicating long-term climate impacts despite net-zero emissions. The results are presented by temporal evolution and by global warming levels, where climatic changes in early and late periods after emission cessation are compared against transient warming under SSP5-8.5. The manuscript provides relevant and novel results such as insights into the climatic effects of delayed emission cessation, regional timing of peak warming after emission cessation, and temperature and precipitation trend differences between transient and stabilized global warming levels.

I do have a few questions and comments, which I consider overall minor, that I would like the authors to address.

*Thank you for your constructive feedback. We believe we can address all your comments as we outline below.*

The manuscript is well written, structured and generally reads fluently. While the manuscript is quite long, the flow of the text does help its understanding. Specific suggestion for sections to be adjusted are listed below.

This study provides a unique dataset as the stabilization simulations are run in emission-driven mode and because they are provided for an entire millennium, which is specifically relevant for 'slow' climate subsystems such as the ocean or cryosphere elements. The authors highlight an important point in line 96, that is relevant for the consideration of climate change experiments for policy under emission budget frameworks. Hence, the experimental design provided here should be inspiration for other modeling groups to at least replicate a sub-ensemble of the experiments described here for a better inter-model comparison.

*Thanks. We also hope that the experimental design is of use to others.*

Given that these simulations are emission driven, I was a bit surprised that no carbon cycle response was analyzed. I understand that ocean and land carbon cycle

responses will affect atmospheric $CO_2$ concentration (which is provided) and therefore its effect on temperature (and precipitation) is shown – ignoring physical feedbacks here, but given the length of these simulations, there would be an opportunity to analyze long-term carbon cycle responses under emission cessation. Since the paper is already long in itself by focusing on temperature and precipitation, I would not expect any additional analysis here, as the authors also refer to further analysis in the last sentence of Section 4. Nonetheless, I would appreciate one or two sentences discussing carbon cycle implications in these emission-driven simulations in the Conclusions Section.

*There is a lot of scope for examining carbon cycle responses in these simulations and there is a follow-up study underway. To do the topic justice and adequately explain the changes we identify we decided a whole other study was needed. We can certainly add a brief discussion of carbon cycle implications of net-zero emissions though.*

Further do I miss a bit a more refined discussion of the results in the frame of other ESM responses (such as under ZECMIP). While the authors make clear that the results can only be interpreted under the specific climate response of ACCESS-ESM-1.5, its differences to other models, or e.g., the ZECMIP multi-model mean, can affect the interpretation of the results presented here. I suggest to either add a few comments in appropriate places in the main text or discuss this issue a bit more in the Conclusions Section.

*We agree that contextualising our findings with reference to multi-model experiments like ZECMIP is useful. We will add additional text to do this.*

An overall limitation of this study is that a single climate model is used, which is very relevant for and interpretation of the result provided here. This is acknowledged by the authors in various places of the main text. However, I think this is a key part to keep in mind when assessing the results of this study. I made specific comments below where I think the manuscript could benefit from some more detail in the framing of the results in the context of other climate models.

*We agree and will make changes in response to your comments below.*

**Specific comments**

From the abstract, I find it is not entire clear whether the focus of the analysis is on the comparison of stabilization vs. transient climate responses, or long-term stabilization at different warming levels only. The reader might appreciate if this can be better clarified.

*There are elements of both. We will clarify the purpose of the study in the revision.*

Line 25 – 'mid-latitude drying trends' – it might be useful for the reader to know here that this refers to extremes provided in the analysis.

*Yes, on reflection this is quite a vague statement that isn't very helpful so we will edit this and perhaps remove this from the Abstract.*

Line 42 – 'near-linear relationship between cumulative carbon dioxide emissions' – there are indications that this might only be true for transient positive emissions.

*Indeed. We can add a clarification of this to make it clear we mean under continued global warming only.*

Line 46 – 'near-zero global mean temperature change for the following century' – that's on a multi-model mean, but individual models can show quite large positive or negative ZEC after emissions cessation.

*Indeed. We can add a point emphasising model uncertainty on this statement.*

Figure 1 – I find it not quite intuitive to present results in panels c) and specifically d) together with the conceptual panels of the experiments in a) and b). I would further like to see a panel showing cumulative emissions in these experiments. I therefore suggest moving panel d) to Figure 3 as the first panel, and exchange it with a panel for cumulative emissions in Figure 1. Also, the panels c) and d) have no x-axis labels, while panels a) and b) do.

*We agree that there are different ways of presenting this information and we think there are merits in different options. We chose figures c and d as approximate parallels to a and b. Admittedly, a and b are drawn to represent a multi-model ensemble and we then plot only data for the ACCESS model in c and d. If we were to plot a cumulative emissions as a new Figure 1d then this would be quite different to Figure 1b. We propose that we largely retain Figure 1 in its current form but add a new Appendix figure with the cumulative emissions graph. We will add axes labels to c and d.*

Lines 115-116 – '… with the starting point chosen…' – As I understand it, I am not convinced by the transferability of this approach to other ESMs because: 1) each ESM will have different long-term climate response – positive or negative ZEC, which vastly affects the branching points from the transient warming experiment, and thus may have different policy implications for when net-zero need to be reached, and 2) reaching temperature targets under non-stable climate response under zero emissions will strongly depend on the intended simulation length, as there is no stabilization (as illustrated in Fig. 1d).

*Indeed, we agree that models would exhibit different responses to net-zero emissions in terms of GMST and other changes. In King et al. (2021) we suggest that prior knowledge of the model could be used to choose different starting points to those that we have*

*applied here for the ACCESS model. We can add some text to clarify this point and make the caveats that you raise.*

Line 150 – '… representation of different ENSO flavors appears to be worse in ACCESS-ESM-1.5 than in other CMIP6 models' – given that a part of the analysis focuses on the ENSO response, I miss some discussion of this point. How does it differ from other ESM? This part could be provided at this point of the main text. And in which way does this affect the results later discussed in the main Results Section and provided in Figure 15? This part could be incorporated in Section 3.4 line 495, where this is already referred to by the authors.

*We agree, some additional discussion would be useful as justification for our analysis. ACCESS-ESM-1.5 has some biases in simulating the spatial pattern of Central and East Pacific ENSO events and lacks seasonality in SST variability in the west-central Pacific seen in observations and some other models. We propose to add some discussion of this point to section 2.3.3 which explains in more detail this model deficiency and explains how we design our analysis in the context of known model biases.*

Line 159 – '… mean surface temperature (GMST) slightly decreases in the first 20-50 years…' – could the authors please add an explanation as to why there is this initial drop?

*Yes, we can add a brief discussion. We believe that the initial decline in temperature is due to the choices we made when transitioning from the SSP5-8.5 run to the stabilised period. At the branching point we set all non-CO2 GHGs back to their pre-industrial values (including aerosols). Some initial analysis has shown that the change in CH4 concentrations back to 1850 levels dominates the initial temperature response, resulting in the observed decrease over the first 50 years. We will include this short explanation in the revised version.*

Line 160 'reduced non-CO2 and aerosol concentrations has taken place…' – The combination of these sentences is slightly confusing as non-CO2 and aerosols were set to 1850 levels as indicated in line 154. Please adjust for clarity.

*Indeed. We will edit this for clarity as we agree this is not currently well worded.*

Lines 157-168 – These lines describe results provided in Figure 1 (see my suggestion on Figure 1 above). I think this paragraph is better placed in the Results Section, e.g. line 281, which would go along nicely with moving Figure 1 panel d) to Figure 3. Please adjust for clarity of the Methods Section.

*In a sense, we are showing initial results here so we agree that this could be moved to the start of the Results section. We propose to move some of the text to the Results section and keep only a brief description of the GMST results here because we would like to*

*retain Figure 2 as a part of the Methods and that requires at least a brief description of GMST changes before we can talk about sampling of GWLs.*

Line 172 – 'quasi-stabilized' – I find this a bit misleading, as emissions are entirely stabilized, but temperatures are clearly not. Perhaps remove.

*Yes, this maybe isn't the best descriptor. We will edit this.*

Section 2.3.1 – This entire section is a listing of analyses that have been performed. This is a bit a question of taste, but I would suggest removing this Section entirely. The reader might have difficulties remembering methodological details when reading about the results later. Rather I would like to see the relevant bits incorporated into the results section where appropriate, which would make following and interpreting the results much easier.

*A similar comment is made in the other review and we agree this is a lengthy section that is perhaps hard to read without also seeing results. We propose to keep a short summary and move some of this text to the Results section accordingly.*

Line 189 – '... with the first period...' – disconnected sentence that sound strange, please revise.

*We agree and will make a suitable edit.*

Line 193 – given that there is often a focus on reaching 1.5°C, the reader might appreciate an extra Figure A3 that is as Figure A2 but keeping NZ2030 as reference (i.e., NZ2035-NZ2030, NZ2040-NZ2030, ...).

*Yes, we agree. We can include this figure the revised version of the paper.*

Figure 2 – the definition of 'early stable' and 'late stable' should be indicated more clearly to be based on *stable emissions,* as temperature are clearly not stable – both in the Figure caption and in the text between lines 220-223.

*Yes, the wording choice here has been challenging, but we agree and will make suitable edits.*

Figure 2 – The reader might appreciate horizontal reference lines for the three warming levels of 1.5, 2 and 3 °C. Also, to me these periods highlighted in bold do not average out at these warming levels, which I would explain by the fact that a range of +/-0.2°C was taken for the definition of the target GWL in these timeseries.

*We agree that this would be useful to include and will add these lines in the revised version as well as a brief explanation of the +/-0.2°C window causing slight differences in averages.*

Figure 2 – If the intention of defining warming levels from the time series was to collect as many years as possible from the available simulation that comply with the +/-0.2°C range around the target GWL, I don't quite understand why simulated period are only chosen from specific simulations. There are clearly periods from other simulations, that reach into this range, but the specific years/decades were not considered here apparently. For example, the simulation branched at 2040 (third grade of orange) drops into the range of temperatures in the bold period of the simulation branched at 2035 (second grade of orange), but the respective years were not considered in the analysis. Such instances can be found for almost all periods highlighted in bold. Could the authors please clarify why this approach has been taken. If it was simply for simplicity of selecting the timeseries for the analysis, this should be made transparent in the main text.

*Yes, we agree that this should be clear. There would be validity to different approaches, but we thought this was a simple way to take advantage of the fact that these runs roughly span the target GWL in the desired early and late time periods. Use of additional runs could inadvertently weight the samples more towards the start or end of the 200-year windows, e.g. using the NZ2035 run to sample the late 1.5°C warming period would result in more weighting nearer 800 than 1000 years post net-zero. This was a minor reason for this choice. We will add some further explanation in the text.*

Lines 224-226 – '…, but results may… GWL ensembles.' – I think this a key part in the assessment and interpretation of the results given the methodology used here. The reader might benefit from the authors providing more detail (or some speculation) on how the described differences may look like and how relevant they are with respect to the results presented here.

*Yes, we agree. We can add some speculative commentary on this point noting this is hard to say confidently what the differences may be.*

Line 227 – 'The method…' – I suggest moving this sentence to line 216.

*Yes, we can make this change.*

Line 228 – 'The use of…' – I suggest moving this sentence to line 218.

*Yes, we can make this change.*

Figure 3 – apart from adding the global mean temperature pane as indicated in the comments above, I would suggest plotting panel a as the land-ocean ratio instead of anomalies. Further, I would plot panels c) and d) as anomalies.

*Thanks for this suggestion. For Figure 3a, the difference rather than ratio is plotted because the ratio is very unstable right at the beginning due to small ocean changes and the fact that 1850-1900 is the baseline and is itself included in the plot. We should*

*explain this choice more because we know others often use the ratio rather than difference. For c) and d) we can make these anomalies in the revised version.*

Line 284 – ', and increase…' – worth noting that is seems to even stabilize at a non-zero difference.

*Yes, we agree and will make a suitable edit.*

Lines 222-223 – '…, suggest that there are complex changes occurring through the next-zero simulations…' – In line with the interpretation provided in lines 341-342, could the authors please comment on – and perhaps incorporate in the main text – whether this is purely because of the regional temperature changes shown in Fig. 5 (i.e. relatively stable Arctic Amplification, and much accelerated Antarctic regional warming due to slow Southern Ocean response), or whether there is indication and/or contribution of other irreversible processes such as non-linear ice sheet responses and feedbacks. Comments in that direction are made in the Conclusions Section, but it might be worth to also discuss this here.

*We can add a comment on this point, although it would be a little speculative too. We would also make the point that more focussed analysis of sea ice changes is needed beyond the overview analysis performed for this paper.*

Lines 471-472 – This sentence is a repetition of lines 462-464.

*Yes, we can rewrite this to make it less repetitive.*

Line 495 – see comment for line 150. It might be good to add some discussion on how ACCESS-ESM-1.5 differs to other models in the context of the ENSO response and how relevant this is for the analysis here.

*Indeed, we agree this is important to include and can briefly comment on this here too.*

**Technical corrections**

*We will make technical corrections to make Figures easier to read and fix and minor typos.*

Axes and/or legend labels are too small in Figures 1, 5, 7, 8, 9, 10, 11, 13, 14, 15. Figure 3 has no x-axis labels, but I assume they would appear too small if they were there, too.

Line 160 – subscript '2' in 'CO2'.

Line 189 – '… temperature difference values …' – add 'between the two periods'.

Line 199 – 'The changing pattern' – remove 'changing'.

Line 305 – '… in the Arctic, also continues…' – remove 'also'.

Line 361 and Line 362 – 'Figure 4X-X' should be 'Figure 7X-X'.

Line 465 – 'G)' should be 'g)'.

Line 503 – 'are not projected' – please check the presence of 'not' in this sentence. For how I understand this sentence, it should be removed.

**Other**

1. Does the paper address relevant scientific questions within the scope of ESD?

   Yes

2. Does the paper present novel concepts, ideas, tools, or data?

   Yes

3. Are substantial conclusions reached?

   Yes

4. Are the scientific methods and assumptions valid and clearly outlined?

   Yes

5. Are the results sufficient to support the interpretations and conclusions?

   Yes

6. Is the description of experiments and calculations sufficiently complete and precise to allow their reproduction by fellow scientists (traceability of results)?

   Yes

7. Do the authors give proper credit to related work and clearly indicate their own new/original contribution?

   Yes

8. Does the title clearly reflect the contents of the paper?

   Yes

9. Does the abstract provide a concise and complete summary?

   Yes

10. Is the overall presentation well structured and clear?

    Yes, see some specific recommendations for the Methods Section

11. Is the language fluent and precise?

    Yes

12. Are mathematical formulae, symbols, abbreviations, and units correctly defined and used?

    Yes

13. Should any parts of the paper (text, formulae, figures, tables) be clarified, reduced, combined, or eliminated?

    Yes, see some specific recommendations for the Methods Section

14. Are the number and quality of references appropriate?

    Yes

15. Is the amount and quality of supplementary material appropriate?

    Yes, with the suggestion to add one panel

---

## Author Comment (AC2)

**General comments**

This paper uses a new set of climate model simulations to explore climate stabilization under zero green gas emissions. This is a very welcome and novel study, which provides an insight into how climate change might evolve, and a useful comparison with many other studies which focus on much shorter simulations with rapidly increasing emissions. The paper has some important findings, for example highlighting that delaying mitigation by even 5 years could have implications for hundreds of years. The paper also illustrates how the impacts of global warming change over time and highlights the need for more research to explore the impacts of different mitigation pathways. The paper is generally fluently written with useful illustrations.

*Thanks for the constructive feedback. We are confident we can address all the comments you outline below.*

I have a few broader comments which might help improve the manuscript:

- I would like to see a bit more discussion of the relevance of these simulations to the real world/ climate policy. The idea that we could instantly stop greenhouse gas emissions is of course hypothetical, and in reality even very strong mitigation would be associated with a decline in emissions over time. This complicates matters as it is more likely to lead to overshoot scenarios, which could have different implications from the scenarios used here. It would be useful to add a short discussion of this.

*Indeed. We agree and we will include additional text, mainly in the Summary and Conclusion section, discussing our results with idealised experiments and their relevance to more realistic emissions pathways.*

- The terminology "net zero" is often used to refer to mitigation which is compliant with the Paris Agreement (i.e. 1.5ºC or 2ºC global warming), whereas in this study some of the net zero scenarios exceed 3º I found this a bit confusing, and I had to keep reminding myself that "net zero" didn't necessarily imply low emissions. I also found it a bit confusing to compare the 3ºC transient sample with the 3ºC stable samples, as I believe they have different cumulative emissions? In some cases a "reduction" (for example in heat extremes) is noted between the transient and stable cases, and it is unclear whether this is a reduction over time in the same scenario, or rather a difference between a rapidly warming 3ºC world with high emissions and stabilized world which has received fewer anthropogenic greenhouse gas emissions and more slowly reached 3ºC. I have added a few comments below that might help make the comparison between these clearer. In addition, I think it would be useful to calculate the cumulative

emissions for each GWL condition. Perhaps this could be added to Table 1? I also wonder whether it might be more straightforward to refer to the scenarios as "zero emission scenarios" rather than "net zero" since I think the scenarios imply that emissions from $CO_2$ sources stop, rather than there being any change in $CO_2$ sinks or $CO_2$ drawdown.

*Thanks. We agree that net-zero emissions pathways are not usually discussed with respect to high global warming levels, so some additional text on this in the Data and Methods section is warranted. We will add clarifying text when discussing the GWL results. We will also add cumulative emissions levels to Table 1 as we agree that it is worth noting that they are different between the transient and stabilising GWL samples. These will be single numbers for the stabilising GWLs and a range for the transient GWLs. We propose to retain "net-zero" nomenclature as this experimental set up is consistent with net-zero carbon dioxide emissions where anthropogenic uptake and drawdown is in balance.*

- The manuscript is generally very well written, but some sections of the results are a bit complicated to read. I have provided line by line comments to help improve clarity.

*Thanks for these comments. We hope to make the text as clear as possible and will make these changes.*

- The paper is quite long, and below you will see I have suggested adding a couple more panels to some of the figures! As a reader I didn't find it too long, but if it needs to be shortened I am not sure if Figure 5 and 6a are both needed. Also, the methods section has quite a bit of detail which I found a little hard to follow without the results, some of the explanation could come during the results figure by figure. The introduction is also quite long, although it is very clearly written and useful.

*Indeed. We agree this is a long paper, but we're pleased you didn't find it too long. We will move some of the Methods description to the Results for clarity (noting the other reviewer made a similar comment). We would like to retain Figures 5 and 6a as we think they provide foundational results relevant to understanding the consequence of stabilisation relative to transient change.*

**Line by line comments**

Line 26 – "differ greatly" – differ with reference to what? Higher greenhouse gas scenarios? Or is this about difference between regions? Please clarify.

*This sentence is about difference between regions. We will clarify this.*

Line 45 – "will result" – suggest to change to "would result" since it is a scenario.

*We agree and will make this edit.*

Line 284 – reference to Joshi et al. 2008. Please make clear how this reference supports – did they find something similar and with what kind of simulation?

*Thanks- we will clarify the use of his reference. Joshi et al. (2008) use simpler GCMs run under different GCM experiments with raised temperatures and $CO_2$ concentrations. These are simpler experiments but they demonstrate a robust reduced land-ocean temperature contrast in equilibrium model simulations relative to transient simulations, although some contrast remains.*

Line 291 – please clarify what you mean by a "fast rate".

*This will be phrased and a range of warming rates will be provided.*

Line 292-295 – suggest to clarify sentence – "even under the lowest global warming simulation broadly aligning with the Paris Agreement" – which simulation do you mean? And, in the latter half of the sentence I think the "beyond" could be removed? Not sure what is meant by "beyond" here.

*This is for the net-zero simulation initialising in 2030 and we agree this should be clearer. We will make a suitable edit and remove the word "beyond".*

Line 299-300 – "relatively small" – relative to the point of zero emissions yes, but there is a change relative to preindustrial. Please clarify.

*We agree and will clarify this.*

Line 303-304 – I think it would be a good idea to put this in the context of the change during the transient simulation. You could say that the models are showing strong persistence in the change in sea ice, since the decrease in sea ice extent experienced during the transient simulation is then broadly maintained (albeit with substantial variability) for 1000 years.

*Yes, we can reframe this following your suggestion.*

Line 305-6 – Do we know why the decline continues after emissions cessation in the Antarctic? Is it due to slower ocean changes in the Antarctic region? It would be nice to comment or perhaps give a hint that you will explore this later in the paper.

*Yes, we discuss this later on, but will edit to make this clearer. We think this is related to the continued warming of the Southern Ocean (at the surface and at depth), but a fuller analysis is needed.*

Line 314-315 – might be helpful to add "over time" for example "Under net-zero emissions, in the Antarctic, there is an increasing change of sea ice free events over time…"

*We agree and will make such an edit.*

Line 330-331 – why might it be related to Southern Ocean warming? Could you give a brief indication (quite interesting!).

*Yes, on reflection this does require some further discussion. There is a suggestion by Oh et al. (2022) that changes to the interhemispheric temperature gradient under increasing and decreasing $CO_2$ concentrations contributes to asymmetric effects on the Indian Monsoon. The reduced interhemispheric gradient in CO2 ramp-down simulations, also seen to a lesser extent in our net-zero emissions simulations, is related to changes in water vapour transport. A brief comment will be added to explain this.*

Line 345-347 – Yes, if the rate of emissions reduction is the same.

*We agree a qualification on this statement is needed to make it clearer.*

Line 356 – "Africa" – more helpful to say "Northern and Central Africa"

*We agree and will make this edit.*

Line 372 – and also dependence on the cumulative emissions? Does the SSP5-8.5 3⁰C have higher cumulative emissions than the late stable 3⁰C?

*Yes, we can add a statement noting this difference.*

Figure 8 and 9 – I wonder whether it would be useful to show and describe the warming patterns for the early stable and late stable periods as well? i.e. the warming relative to preindustrial rather than the difference between SSP5-8.5 and the stable periods? It might be nice to make the point clearly that a 3⁰C "transient" world looks very different from a 3⁰C "stabilized" world – i.e. (I think) – 3⁰C transient has huge warming over continents and some warming over sea ice regions, 3⁰C transient has more uniform warming? Are the continents showing warming similar to the global mean of 3⁰C? As a reader I'd quite like to visualize this.

*Yes, we agree. We can add these as additional figures to the Appendices. Adding these maps to Figures 8 and 9 would make them too crowded we think.*

418-419 – "suggest very large areas of the world may exhibit some return towards pre-industrial levels of seasonal-average precipitation." – very interesting. Again, could you show this? Even as an appendix.

*Yes, we agree this is an interesting implication of Figures 10-12. We can add a new Figure showing the areas of the world where precipitation reversal means changes in seasonal distributions are statistically indistinguishable from the pre-industrial period when they*

*were significantly different during the transient period. This could be as an addition to Figure 12 or as new Appendix Figures depending on the findings.*

480 – "marked reduction" – is there a decrease over time as the climate stabilizes, or is this because a different simulation is used for the "early stable 1.5⁰C" vs the "late stable 1.5⁰C". If the extremes actually decrease over time, despite the cumulative emissions being the same, this is really interesting and could be highlighted more. Either way it would be nice to comment on this.

*This is for a decrease in hot extremes over land areas for a given GWL between initial and later stabilisation periods. These extremes are drawn from different simulations with different cumulative emissions levels at the same GWL. We will add a statement noting this to make it clearer.*

500-505 – Please could you clarify whether you expect that the stabilization scenarios show a decrease after emissions cessation? Or is the reduction between the 21ˢᵗ century and the net zero simulations shown in Figure 15 a result of the different emissions (where this 21ˢᵗ century simulation includes emissions and temperatures which exceed any of the net zero stabilization scenarios?)

*Thanks. We will clarify this. We don't see significant differences in ENSO amplitude or frequency between different samples from the net-zero simulations. The difference arises between the 21ˢᵗ century rapid warming and stabilisation periods. We will explain this more clearly.*

522 – "reduced land temperature means and extremes" – is there a reduction over time? Or is it a reduction compared to a transient world with the same GWL (and higher emissions)?

*This is over time, but we will edit to clarify this.*

Figure 14 caption – check references to figure letters. I think "e, f" should be "d, e"

*Thanks for spotting this typo. We will make this correction.*

1. Does the paper address relevant scientific questions within the scope of ESD? Yes
2. Does the paper present novel concepts, ideas, tools, or data? Yes
3. Are substantial conclusions reached? Yes
4. Are the scientific methods and assumptions valid and clearly outlined? Yes
5. Are the results sufficient to support the interpretations and conclusions? Yes

6. Is the description of experiments and calculations sufficiently complete and precise to allow their reproduction by fellow scientists (traceability of results)? Yes

7. Do the authors give proper credit to related work and clearly indicate their own new/original contribution? Yes

8. Does the title clearly reflect the contents of the paper? Yes

9. Does the abstract provide a concise and complete summary? Yes

10. Is the overall presentation well structured and clear? Yes

11. Is the language fluent and precise? Yes, generally, I've made suggestions for places where it could be clearer.

12. Are mathematical formulae, symbols, abbreviations, and units correctly defined and used? Yes

13. Should any parts of the paper (text, formulae, figures, tables) be clarified, reduced, combined, or eliminated? Yes

14. Are the number and quality of references appropriate? Yes

15. Is the amount and quality of supplementary material appropriate? Yes

---

## Author Response (AR1)

**Reply to Reviewers' comments**

**Reply to Rachel James's comment**

**General comments**

This paper uses a new set of climate model simulations to explore climate stabilization under zero green gas emissions. This is a very welcome and novel study, which provides an insight into how climate change might evolve, and a useful comparison with many other studies which focus on much shorter simulations with rapidly increasing emissions. The paper has some important findings, for example highlighting that delaying mitigation by even 5 years could have implications for hundreds of years. The paper also illustrates how the impacts of global warming change over time and highlights the need for more research to explore the impacts of different mitigation pathways. The paper is generally fluently written with useful illustrations.

*Thanks for the constructive feedback. We believe we have addressed all the comments you outline below.*

I have a few broader comments which might help improve the manuscript:

- I would like to see a bit more discussion of the relevance of these simulations to the real world/ climate policy. The idea that we could instantly stop greenhouse gas emissions is of course hypothetical, and in reality even very strong mitigation would be associated with a decline in emissions over time. This complicates matters as it is more likely to lead to overshoot scenarios, which could have different implications from the scenarios used here. It would be useful to add a short discussion of this.

*Indeed. We agree and we have included additional text, in the Methods, and the Summary and Conclusion sections, discussing our results with idealised experiments and the need for more realistic emissions pathways.*

- The terminology "net zero" is often used to refer to mitigation which is compliant with the Paris Agreement (i.e. 1.5ºC or 2ºC global warming), whereas in this study some of the net zero scenarios exceed 3º I found this a bit confusing, and I had to keep reminding myself that "net zero" didn't necessarily imply low emissions. I also found it a bit confusing to compare the 3ºC transient sample with the 3ºC stable samples, as I believe they have different cumulative emissions? In some cases a "reduction" (for example in heat extremes) is noted between the transient and stable cases, and it is unclear whether this is a reduction over time in the same scenario, or rather a difference between a rapidly warming 3ºC world with high emissions and stabilized world which has received fewer anthropogenic greenhouse gas emissions and more slowly reached 3ºC. I have added a

few comments below that might help make the comparison between these clearer. In addition, I think it would be useful to calculate the cumulative emissions for each GWL condition. Perhaps this could be added to Table 1? I also wonder whether it might be more straightforward to refer to the scenarios as "zero emission scenarios" rather than "net zero" since I think the scenarios imply that emissions from $CO_2$ sources stop, rather than there being any change in $CO_2$ sinks or $CO_2$ drawdown.

*Thanks. We agree that net-zero emissions pathways are not usually discussed with respect to high global warming levels, so some additional text has been added to section 2.2. We have also added cumulative emissions levels to Table 1 as we agree that it is worth noting that they are different between the transient and stabilising GWL samples. These are single numbers for the stabilising GWLs and a range for the transient GWLs. We retain "net-zero" nomenclature as this experimental set up is consistent with net-zero carbon dioxide emissions where anthropogenic uptake and drawdown is in balance.*

- The manuscript is generally very well written, but some sections of the results are a bit complicated to read. I have provided line by line comments to help improve clarity.

*Thanks for these comments. We hope to make the text as clear as possible and have made these changes.*

- The paper is quite long, and below you will see I have suggested adding a couple more panels to some of the figures! As a reader I didn't find it too long, but if it needs to be shortened I am not sure if Figure 5 and 6a are both needed. Also, the methods section has quite a bit of detail which I found a little hard to follow without the results, some of the explanation could come during the results figure by figure. The introduction is also quite long, although it is very clearly written and useful.

*Indeed. We agree this is a long paper, but we're pleased you didn't find it too long. We have moved some of the Methods description to the Results for clarity. We have retained Figures 5 and 6a as we think they provide foundational results relevant to understanding the consequence of stabilisation relative to transient change.*

**Line by line comments**

Line 26 – "differ greatly" – differ with reference to what? Higher greenhouse gas scenarios? Or is this about difference between regions? Please clarify.

*This sentence is about difference between regions. We have clarified this.*

Line 45 – "will result" – suggest to change to "would result" since it is a scenario.

*We agree and have made this edit.*

Line 284 – reference to Joshi et al. 2008. Please make clear how this reference supports – did they find something similar and with what kind of simulation?

*Thanks- we have clarified the use of this reference. Joshi et al. (2008) use simpler GCMs run under different GCM experiments with raised temperatures and $CO_2$ concentrations. These are simpler experiments but they demonstrate a robust reduced land-ocean temperature contrast in equilibrium model simulations relative to transient simulations, although some contrast remains.*

Line 291 – please clarify what you mean by a "fast rate".

*We have rephrased this sentence to avoid using this qualitative language.*

Line 292-295 – suggest to clarify sentence – "even under the lowest global warming simulation broadly aligning with the Paris Agreement" – which simulation do you mean? And, in the latter half of the sentence I think the "beyond" could be removed? Not sure what is meant by "beyond" here.

*This is for the net-zero simulation initialising in 2030. We have made edits to make this clearer.*

Line 299-300 – "relatively small" – relative to the point of zero emissions yes, but there is a change relative to preindustrial. Please clarify.

*We agree and have clarified this.*

Line 303-304 – I think it would be a good idea to put this in the context of the change during the transient simulation. You could say that the models are showing strong persistence in the change in sea ice, since the decrease in sea ice extent experienced during the transient simulation is then broadly maintained (albeit with substantial variability) for 1000 years.

*Yes, we have reframed this discussion.*

Line 305-6 – Do we know why the decline continues after emissions cessation in the Antarctic? Is it due to slower ocean changes in the Antarctic region? It would be nice to comment or perhaps give a hint that you will explore this later in the paper.

*We have added text later on that discusses this in more detail with reference to the strong Southern Ocean warming after emissions cessation.*

Line 314-315 – might be helpful to add "over time" for example "Under net-zero emissions, in the Antarctic, there is an increasing change of sea ice free events over time…"

*We agree and have made this edit.*

Line 330-331 – why might it be related to Southern Ocean warming? Could you give a brief indication (quite interesting!).

*Yes, on reflection this does require some further discussion. There is a suggestion by Oh et al. (2022) that changes to the interhemispheric temperature gradient under increasing and decreasing $CO_2$ concentrations contributes to asymmetric effects on the Indian Monsoon. The reduced interhemispheric gradient in CO2 ramp-down simulations, also seen to a lesser extent in our net-zero emissions simulations, is related to changes in water vapour transport. A brief comment has been added to explain this.*

Line 345-347 – Yes, if the rate of emissions reduction is the same.

*We have added a qualification on this paragraph.*

Line 356 – "Africa" – more helpful to say "Northern and Central Africa"

*We have made this edit.*

Line 372 – and also dependence on the cumulative emissions? Does the SSP5-8.5 3⁰C have higher cumulative emissions than the late stable 3⁰C?

*Yes, it does as we now note in Table 1. We think raising this here might be confusing though as there are many different ways in which different cumulative emissions could be reached and we are only comparing a small sample of these.*

Figure 8 and 9 – I wonder whether it would be useful to show and describe the warming patterns for the early stable and late stable periods as well? i.e. the warming relative to preindustrial rather than the difference between SSP5-8.5 and the stable periods? It might be nice to make the point clearly that a 3⁰C "transient" world looks very different from a 3⁰C "stabilized" world – i.e. (I think) – 3⁰C transient has huge warming over continents and some warming over sea ice regions, 3⁰C transient has more uniform warming? Are the continents showing warming similar to the global mean of 3⁰C? As a reader I'd quite like to visualize this.

*Yes, we agree. We have added these as additional figures to the Appendices (new Figures A5, A6).*

418-419 – "suggest very large areas of the world may exhibit some return towards pre-industrial levels of seasonal-average precipitation." – very interesting. Again, could you show this? Even as an appendix.

*Yes, we agree this is an interesting implication of Figures 10-12. We have found that this return to pre-industrial levels of precipitation applies to small areas of the world (<10%). Rather than cluttering Figure 12, we have added an Appendix (new Figure A7) which shows this.*

480 – "marked reduction" – is there a decrease over time as the climate stabilizes, or is this because a different simulation is used for the "early stable 1.5⁰C" vs the "late stable 1.5⁰C". If the extremes actually decrease over time, despite the cumulative emissions being the same, this is really interesting and could be highlighted more. Either way it would be nice to comment on this.

*This is for a decrease in hot extremes over land areas for a given GWL between initial and later stabilisation periods. These extremes are drawn from different simulations with different cumulative emissions levels at the same GWL. We have rephrased this to make it clearer.*

500-505 – Please could you clarify whether you expect that the stabilization scenarios show a decrease after emissions cessation? Or is the reduction between the 21st century and the net zero simulations shown in Figure 15 a result of the different emissions (where this 21st century simulation includes emissions and temperatures which exceed any of the net zero stabilization scenarios?)

*Thanks. We have sought to clarify this. We don't see significant differences in ENSO amplitude or frequency between different samples from the net-zero simulations. The difference arises between the 21st century rapid warming and stabilisation periods.*

522 – "reduced land temperature means and extremes" – is there a reduction over time? Or is it a reduction compared to a transient world with the same GWL (and higher emissions)?

*This is more between GWLs so we have rephrased this to be over time and significantly modified the sentence as a result.*

Figure 14 caption – check references to figure letters. I think "e, f" should be "d, e"

*Thanks for spotting this typo. We have made this correction.*

1. Does the paper address relevant scientific questions within the scope of ESD? Yes

2. Does the paper present novel concepts, ideas, tools, or data? Yes

3.  Are substantial conclusions reached? Yes

4.  Are the scientific methods and assumptions valid and clearly outlined? Yes

5.  Are the results sufficient to support the interpretations and conclusions? Yes

6.  Is the description of experiments and calculations sufficiently complete and precise to allow their reproduction by fellow scientists (traceability of results)? Yes

7.  Do the authors give proper credit to related work and clearly indicate their own new/original contribution? Yes

8.  Does the title clearly reflect the contents of the paper? Yes

9.  Does the abstract provide a concise and complete summary? Yes

10. Is the overall presentation well structured and clear? Yes

11. Is the language fluent and precise? Yes, generally, I've made suggestions for places where it could be clearer.

12. Are mathematical formulae, symbols, abbreviations, and units correctly defined and used? Yes

13. Should any parts of the paper (text, formulae, figures, tables) be clarified, reduced, combined, or eliminated? Yes

14. Are the number and quality of references appropriate? Yes

15. Is the amount and quality of supplementary material appropriate? Yes

**Reply to Norman Steinert's comment**

**General comments**

This manuscript analyzes a new set of climate change experiments utilizing the ACCESS-ESM-1.5 Earth system model to simulate climate evolution under net-zero emissions. Seven simulations spanning 1000 years were branched of the SSP5-8.5 simulation at 5-year intervals starting in 2030 to assess climate stabilization in line with the Paris Agreement and higher warming scenarios. Significant findings are provided for continued Southern Ocean warming, reversal of mid-latitude drying trends, and varying sea ice extent trajectories in the Arctic and Antarctic. The study also examines the El Niño-Southern Oscillation (ENSO), noting reduced amplitude and frequency of events under climate stabilization. The results show that regional changes persist for centuries post-emission cessation, indicating long-term climate impacts despite net-zero emissions. The results are presented by temporal evolution and by global warming levels, where climatic changes in early and late periods after emission cessation are compared against transient warming under SSP5-8.5. The manuscript provides relevant and novel results such as insights into the climatic effects of delayed emission cessation, regional timing of peak warming after emission cessation, and temperature and precipitation trend differences between transient and stabilized global warming levels.

I do have a few questions and comments, which I consider overall minor, that I would like the authors to address.

*Thank you for your constructive feedback. We have sought to address all of your comments as we outline below.*

The manuscript is well written, structured and generally reads fluently. While the manuscript is quite long, the flow of the text does help its understanding. Specific suggestion for sections to be adjusted are listed below.

This study provides a unique dataset as the stabilization simulations are run in emission-driven mode and because they are provided for an entire millennium, which is specifically relevant for 'slow' climate subsystems such as the ocean or cryosphere elements. The authors highlight an important point in line 96, that is relevant for the consideration of climate change experiments for policy under emission budget frameworks. Hence, the experimental design provided here should be inspiration for other modeling groups to at least replicate a sub-ensemble of the experiments described here for a better inter-model comparison.

*Thanks. We also hope that the experimental design is of use to others.*

Given that these simulations are emission driven, I was a bit surprised that no carbon cycle response was analyzed. I understand that ocean and land carbon cycle

responses will affect atmospheric $CO_2$ concentration (which is provided) and therefore its effect on temperature (and precipitation) is shown – ignoring physical feedbacks here, but given the length of these simulations, there would be an opportunity to analyze long-term carbon cycle responses under emission cessation. Since the paper is already long in itself by focusing on temperature and precipitation, I would not expect any additional analysis here, as the authors also refer to further analysis in the last sentence of Section 4. Nonetheless, I would appreciate one or two sentences discussing carbon cycle implications in these emission-driven simulations in the Conclusions Section.

*There is a lot of scope for examining carbon cycle responses in these simulations and there is a follow-up study underway. To do the topic justice and adequately explain the changes we identify we decided a whole other study was needed. We have added a couple of additional sentences in the Conclusions on this topic though.*

Further do I miss a bit a more refined discussion of the results in the frame of other ESM responses (such as under ZECMIP). While the authors make clear that the results can only be interpreted under the specific climate response of ACCESS-ESM-1.5, its differences to other models, or e.g., the ZECMIP multi-model mean, can affect the interpretation of the results presented here. I suggest to either add a few comments in appropriate places in the main text or discuss this issue a bit more in the Conclusions Section.

*We agree that contextualising our findings with reference to multi-model experiments like ZECMIP is useful. We have added to the Conclusions section a sentence on this.*

An overall limitation of this study is that a single climate model is used, which is very relevant for and interpretation of the result provided here. This is acknowledged by the authors in various places of the main text. However, I think this is a key part to keep in mind when assessing the results of this study. I made specific comments below where I think the manuscript could benefit from some more detail in the framing of the results in the context of other climate models.

*We agree and will make changes in response to your comments below.*

**Specific comments**

From the abstract, I find it is not entire clear whether the focus of the analysis is on the comparison of stabilization vs. transient climate responses, or long-term stabilization at different warming levels only. The reader might appreciate if this can be better clarified.

*There are elements of both which we hope we have clarified with additional text in the Abstract that adds emphasis on effects of different GWLs.*

Line 25 – 'mid-latitude drying trends' – it might be useful for the reader to know here that this refers to extremes provided in the analysis.

*We have edited this to more simply say that some regions experience changes in precipitation trends.*

Line 42 – 'near-linear relationship between cumulative carbon dioxide emissions' – there are indications that this might only be true for transient positive emissions.

*Indeed. We have added a clarification of this to make it clear we mean in a transient climate state specifically.*

Line 46 – 'near-zero global mean temperature change for the following century' – that's on a multi-model mean, but individual models can show quite large positive or negative ZEC after emissions cessation.

*Indeed. We have added a point emphasising model uncertainty on this statement.*

Figure 1 – I find it not quite intuitive to present results in panels c) and specifically d) together with the conceptual panels of the experiments in a) and b). I would further like to see a panel showing cumulative emissions in these experiments. I therefore suggest moving panel d) to Figure 3 as the first panel, and exchange it with a panel for cumulative emissions in Figure 1. Also, the panels c) and d) have no x-axis labels, while panels a) and b) do.

*We agree that there are different ways of presenting this information and we think there are merits in different options. We chose figures c and d as approximate parallels to a and b. Admittedly, a and b are drawn to represent a multi-model ensemble and we then plot only data for the ACCESS model in c and d. If we were to plot a cumulative emissions as a new Figure 1d then this would be quite different to Figure 1b. We have largely retained Figure 1 in its current form but added a new Appendix figure (Figure A1) with the cumulative emissions graph. We have added axes labels to c and d.*

Lines 115-116 – '… with the starting point chosen…' – As I understand it, I am not convinced by the transferability of this approach to other ESMs because: 1) each ESM will have different long-term climate response – positive or negative ZEC, which vastly affects the branching points from the transient warming experiment, and thus may have different policy implications for when net-zero need to be reached, and 2) reaching temperature targets under non-stable climate response under zero emissions will strongly depend on the intended simulation length, as there is no stabilization (as illustrated in Fig. 1d).

*Indeed, we agree that models would exhibit different responses to net-zero emissions in terms of GMST and other changes. In King et al. (2021) we suggest that prior knowledge of the model could be used to choose different starting points to those that we have applied here for the ACCESS model. We have added some text to clarify this point.*

Line 150 – '... representation of different ENSO flavors appears to be worse in ACCESS-ESM-1.5 than in other CMIP6 models' – given that a part of the analysis focuses on the ENSO response, I miss some discussion of this point. How does it differ from other ESM? This part could be provided at this point of the main text. And in which way does this affect the results later discussed in the main Results Section and provided in Figure 15? This part could be incorporated in Section 3.4 line 495, where this is already referred to by the authors.

*We agree, some additional discussion is useful as justification for our analysis. ACCESS-ESM-1.5 has some biases in simulating the spatial pattern of Central and East Pacific ENSO events and lacks seasonality in SST variability in the west-central Pacific seen in observations and some other models. We add some explanation of the model deficiency in section 2.1 and explain the rationale behind the analysis further in section 3.4.*

Line 159 – '... mean surface temperature (GMST) slightly decreases in the first 20-50 years...' – could the authors please add an explanation as to why there is this initial drop?

*Yes, we can add a brief discussion. We believe that the initial decline in temperature is due to the choices we made when transitioning from the SSP5-8.5 run to the stabilised period. At the branching point we set all non-CO2 GHGs back to their pre-industrial values (including aerosols). Some initial analysis has shown that the change in CH4 concentrations back to 1850 levels dominates the initial temperature response, resulting in the observed decrease over the first 50 years. We have included this short explanation in the revised version.*

Line 160 'reduced non-CO2 and aerosol concentrations has taken place...' – The combination of these sentences is slightly confusing as non-CO2 and aerosols were set to 1850 levels as indicated in line 154. Please adjust for clarity.

*Indeed. We have edited this for clarity as we agree this was not well worded.*

Lines 157-168 – These lines describe results provided in Figure 1 (see my suggestion on Figure 1 above). I think this paragraph is better placed in the Results Section, e.g. line 281, which would go along nicely with moving Figure 1 panel d) to Figure 3. Please adjust for clarity of the Methods Section.

*In a sense, we are showing initial results here so we agree that this could be moved to the start of the Results section. We have moved some of the text to the Results section and keep only a brief description of the GMST results here because we would like to retain Figure 2 as a part of the Methods and that requires at least a brief description of GMST changes before we can talk about sampling of GWLs.*

Line 172 – 'quasi-stabilized' – I find this a bit misleading, as emissions are entirely stabilized, but temperatures are clearly not. Perhaps remove.

*Yes, we have removed this.*

Section 2.3.1 – This entire section is a listing of analyses that have been performed. This is a bit a question of taste, but I would suggest removing this Section entirely. The reader might have difficulties remembering methodological details when reading about the results later. Rather I would like to see the relevant bits incorporated into the results section where appropriate, which would make following and interpreting the results much easier.

*A similar comment is made in the other review and we agree this is a lengthy section that is perhaps hard to read without also seeing results. We have shortened section 2.3.1 particularly and we hope this improves readability.*

Line 189 – '... with the first period...' – disconnected sentence that sound strange, please revise.

*This sentence has been removed following edits with respect to the previous comment.*

Line 193 – given that there is often a focus on reaching 1.5°C, the reader might appreciate an extra Figure A3 that is as Figure A2 but keeping NZ2030 as reference (i.e., NZ2035-NZ2030, NZ2040-NZ2030, ...).

*Yes, we agree. We have included this figure the revised version of the paper (new Figure A4).*

Figure 2 – the definition of 'early stable' and 'late stable' should be indicated more clearly to be based on *stable emissions,* as temperature are clearly not stable – both in the Figure caption and in the text between lines 220-223.

*We have made minor to the Figure caption and subsequent text that try and clarify this.*

Figure 2 – The reader might appreciate horizontal reference lines for the three warming levels of 1.5, 2 and 3 °C. Also, to me these periods highlighted in bold do

not average out at these warming levels, which I would explain by the fact that a range of +/-0.2°C was taken for the definition of the target GWL in these timeseries.

*We agree that this would be useful to include and have added these lines in the revised version as well as a brief explanation of the +/-0.2°C window causing slight differences in averages.*

Figure 2 – If the intention of defining warming levels from the time series was to collect as many years as possible from the available simulation that comply with the +/-0.2°C range around the target GWL, I don't quite understand why simulated period are only chosen from specific simulations. There are clearly periods from other simulations, that reach into this range, but the specific years/decades were not considered here apparently. For example, the simulation branched at 2040 (third grade of orange) drops into the range of temperatures in the bold period of the simulation branched at 2035 (second grade of orange), but the respective years were not considered in the analysis. Such instances can be found for almost all periods highlighted in bold. Could the authors please clarify why this approach has been taken. If it was simply for simplicity of selecting the timeseries for the analysis, this should be made transparent in the main text.

*Yes, we agree that this should be clear. There would be validity to different approaches, but we thought this was a simple way to take advantage of the fact that these runs roughly span the target GWL in the desired early and late time periods. Use of additional runs could inadvertently weight the samples more towards the start or end of the 200-year windows, e.g. using the NZ2035 run to sample the late 1.5°C warming period would result in more weighting nearer 800 than 1000 years post net-zero. This was a minor reason for this choice. We have added some further explanation in the text.*

Lines 224-226 – '..., but results may... GWL ensembles.' – I think this a key part in the assessment and interpretation of the results given the methodology used here. The reader might benefit from the authors providing more detail (or some speculation) on how the described differences may look like and how relevant they are with respect to the results presented here.

*This is an interesting point. We agree this is an important aspect of the method, but on reflection we're not confident of what difference this would make, and it feels too speculative to comment. We note this is different from the suggested additional text in the initial response to reviewer comments.*

Line 227 – 'The method...' – I suggest moving this sentence to line 216.

*Yes, we have made this change.*

Line 228 – 'The use of...' – I suggest moving this sentence to line 218.

*Yes, we have made this change.*

Figure 3 – apart from adding the global mean temperature pane as indicated in the comments above, I would suggest plotting panel a as the land-ocean ratio instead of anomalies. Further, I would plot panels c) and d) as anomalies.

*Thanks for this suggestion. For Figure 3a, the difference rather than ratio is plotted because the ratio is very unstable right at the beginning due to small ocean changes and the fact that 1850-1900 is the baseline and is itself included in the plot. We should explain this choice more because we know others often use the ratio rather than difference. For c) and d) we have made these anomalies in the revised version.*

Line 284 – ', and increase…' – worth noting that is seems to even stabilize at a non-zero difference.

*Yes, we agree and have noted this in the revised text.*

Lines 222-223 – '…, suggest that there are complex changes occurring through the next-zero simulations…' – In line with the interpretation provided in lines 341-342, could the authors please comment on – and perhaps incorporate in the main text – whether this is purely because of the regional temperature changes shown in Fig. 5 (i.e. relatively stable Arctic Amplification, and much accelerated Antarctic regional warming due to slow Southern Ocean response), or whether there is indication and/or contribution of other irreversible processes such as non-linear ice sheet responses and feedbacks. Comments in that direction are made in the Conclusions Section, but it might be worth to also discuss this here.

*We have added a comment on this point. We think that more focussed analysis of sea ice changes is needed beyond the overview analysis performed for this paper.*

Lines 471-472 – This sentence is a repetition of lines 462-464.

*We agree and have made edits to reduce repetition.*

Line 495 – see comment for line 150. It might be good to add some discussion on how ACCESS-ESM-1.5 differs to other models in the context of the ENSO response and how relevant this is for the analysis here.

*We have extended the text here to better justify our methods.*

**Technical corrections**

*We have corrected all typos listed here except on the line 503 comment where the sentence was correct. We have edited it to make it clearer though.*

***Figures have been modified slightly with additional axes labels and enlarged fonts. Note, that .eps figures will be supplied which will be clearer than these ones. The size of the figure files prohibits having very high-resolution versions in the word document submission.***

Axes and/or legend labels are too small in Figures 1, 5, 7, 8, 9, 10, 11, 13, 14, 15. Figure 3 has no x-axis labels, but I assume they would appear too small if they were there, too.

Line 160 – subscript '2' in 'CO2'.

Line 189 – '… temperature difference values …' – add 'between the two periods'.

Line 199 – 'The changing pattern' – remove 'changing'.

Line 305 – '… in the Arctic, also continues…' – remove 'also'.

Line 361 and Line 362 – 'Figure 4X-X' should be 'Figure 7X-X'.

Line 465 – 'G)' should be 'g)'.

Line 503 – 'are not projected' – please check the presence of 'not' in this sentence. For how I understand this sentence, it should be removed.

**Other**

1. Does the paper address relevant scientific questions within the scope of ESD?

   Yes

2. Does the paper present novel concepts, ideas, tools, or data?

   Yes

3. Are substantial conclusions reached?

   Yes

4. Are the scientific methods and assumptions valid and clearly outlined?

   Yes

5. Are the results sufficient to support the interpretations and conclusions?

   Yes

6. Is the description of experiments and calculations sufficiently complete and precise to allow their reproduction by fellow scientists (traceability of results)?

   Yes

7. Do the authors give proper credit to related work and clearly indicate their own new/original contribution?

   Yes

8. Does the title clearly reflect the contents of the paper?

   Yes

9. Does the abstract provide a concise and complete summary?

   Yes

10. Is the overall presentation well structured and clear?

    Yes, see some specific recommendations for the Methods Section

11. Is the language fluent and precise?

    Yes

12. Are mathematical formulae, symbols, abbreviations, and units correctly defined and used?

    Yes

13. Should any parts of the paper (text, formulae, figures, tables) be clarified, reduced, combined, or eliminated?

    Yes, see some specific recommendations for the Methods Section

14. Are the number and quality of references appropriate?

    Yes

15. Is the amount and quality of supplementary material appropriate?

    Yes, with the suggestion to add one panel

---

## Author Response (AR2)

**Response to reviewers**

*We thank the reviewers for their constructive feedback and helpful comments which we've used to try and improve the clarity of the manuscript. Please find line-by-line responses to each comment below in red and italics.*

Reviewer: Norman Steinert

The authors have done a good job incorporating the suggestions and clarifications.

Please check the first sentence in line 236 (270) of the revised manuscript (with track changes), which either looks incomplete or out of place.

Other than that, I have no further comments on the current version.

*Thank you. Yes, we agree and have decided to simply remove this sentence (L238).*

Reviewer: Rachel James

The authors have responded thoroughly to the comments from me and the other reviewer. I think the manuscript is largely ready for publication.

*Thank you.*

One of my comments was "I ... found it a bit confusing to compare the $3^0$C transient sample with the $3^0$C stable samples, as I believe they have different cumulative emissions? In some cases a "reduction" (for example in heat extremes) is noted between the transient and stable cases, and it is unclear whether this is a reduction over time in the same scenario, or rather a difference between a rapidly warming $3^0$C world with high emissions and stabilized world which has received fewer anthropogenic greenhouse gas emissions and more slowly reached $3^0$C."

The authors have responded to this by clarifying the text. However, I have still noticed a couple of places where it is a bit unclear. I suggest that the authors check these and clarify:

1. Figure 12 figure caption refers to "reversal" - is "reversal" the right word, since it is comparing two different simulations? Is there evidence that there is a change in the direction of the trend over time? Or is it a difference between a transient state and a (differently forced) stable state. Please clarify in the figure caption.

2. Description of Figure 12 refers to "return" and "reversal" - is there evidence for a return? Or is more a difference between two different experiments?

3. Figure A7 caption and description - again, is "reversal" accurate? Please clarify.

*Thanks for these points. Given they're all along the same theme we respond to them together. We see your point that we are comparing precipitation changes between different simulations which aren't completely sequential, so care needs to be taken in describing these results. We have made edits in the locations in the text that you point to (L445-463, 627-628), as well as the Data and Methods section (L246, 250) so that we don't refer to reversal anymore. Rather we refer to locations where the trends are significantly different and change depending on examining transient or stabilised climate projections.*